# Identification of an embryonic differentiation stage marked by Sox1 and FoxA2 co-expression using combined cell tracking and high dimensional protein imaging

Geethika Arekatla[1,4,5], Stavroula Skylaki[1], David Corredor Suarez[1], Hartland Jackson [2,3,6], Denis Schapiro [2,3,7,8,9], Stefanie Engler[2], Markus Auler[1], German Camargo Ortega [1], Simon Hastreiter[1], Andreas Reimann[1], Dirk Loeffler[1,10,11], Bernd Bodenmiller [2,3] & Timm Schroeder [1] ✉

Pluripotent mouse embryonic stem cells (ESCs) can differentiate to all germ layers and serve as an in vitro model of embryonic development. To better understand the differentiation paths traversed by ESCs committing to different lineages, we track individual differentiating ESCs by timelapse imaging followed by multiplexed high-dimensional Imaging Mass Cytometry (IMC) protein quantification. This links continuous live single-cell molecular NANOG and cellular dynamics quantification over 5-6 generations to protein expression of 37 different molecular regulators in the same single cells at the observation endpoints. Using this unique data set including kinship history and live lineage marker detection, we show that NANOG downregulation occurs generations prior to, but is not sufficient for neuroectoderm marker Sox1 upregulation. We identify a developmental cell type co-expressing both the canonical Sox1 neuroectoderm and FoxA2 endoderm markers in vitro and confirm the presence of such a population in the post-implantation embryo. RNASeq reveals cells co-expressing SOX1 and FOXA2 to have a unique cell state characterized by expression of both endoderm as well as neuroectoderm genes suggesting lineage potential towards both germ layers.

ESCs are a commonly used model to interrogate embryonic cell lineage choices. Derived from the inner cell mass of mouse embryos and kept in a pluripotent state in vitro, they are an accessible in vitro model of development in controlled conditions[1]. Numerous studies have revealed transcription factors, signaling molecules and epigenetic molecules that are implicated in ESC pluripotency and differentiation networks[2–5]. We, thus, have a good understanding of the molecular signatures of initial and terminal cell fates as well as the identity of individual molecular regulators.

However, we still lack information of the precise transition steps that individual cells traverse from pluripotency to one of the three germ layer lineages. To quantify the dynamics of fate decisions, we need to follow individual differentiating cells over time with continuous information of the cells' state at each timepoint[6–14]. So far, one

**Fig. 1 | Combined timelapse and Imaging Mass Cytometry of differentiating ESCs reveals heterogeneous protein expression even between closely related individual cells in the same media conditions. A** Experimental workflow. **B** Imaging data reveals heterogeneous protein expression at the single cell level. Representative example in RA medium. Nucleus labelled by iRFPnucmem in timelapse images. Scale bar 50 um. Lineage data and NANOGVenus quantification over 46 h and 5 generations. Branches terminating before 46 h due to cell death (X) also shown. Corresponding cell and progeny (pink circles) highlighted in primary imaging data. See also Supplementary Fig. 1B for complete field of view of this zoomed-in image. **C** Quantification of protein expression in different media conditions obtained by IMC at 46 h, plotted as distribution of expression profile of each protein per condition n > 9000 cells per media condition from 2 biological replicates. **D** UMAP dimensionality reduction fails to separate cells into distinct fate marker enriched populations. Cells colored by culture media (top left) or antibody expression levels (rest). Dimensions used include all 37 protein markers. n > 34000 cells from 2 biological replicates. **E** Representative example of thresholding proteins upregulated upon differentiation (SOX1 from IMC quantification). Threshold value (pink) and % SOX1+ cells in per condition. n > 9000 cells per condition from two biological replicates. **F** Representative example of threshold determination for proteins downregulated during differentiation (NANOG from IMC). Bimodal expression fitted with gamma distributions (yellow) and threshold value (pink) at intersection of the two distributions. Proportions of cells in NANOG+ class per condition as indicated. n > 9000 cells per condition from two biological replicates.

addressed by recent studies combining cell tracking with end point single cell sequencing[20–23], which however do not provide information about protein expression, subcellular localization, and posttranslational protein modifications important for their regulation.

Here, we performed timelapse microscopy and single-cell tracking of single differentiating ESCs[8,24,25] followed by imaging mass cytometry (IMC)-based protein quantification to acquire a comprehensive data set consisting of both cellular and molecular dynamics history over 5 to 6 generations and high dimensional information on molecular regulator expression in the final cell states. Making use of this combined information, we investigate the link between protein and cellular dynamics, such as morphology, motility, division time, and final cell fate. We identify a developmental cell population co-expressing SOX1 and FOXA2, confirm its presence in-vivo; and find that FOXA2, an endoderm marker, can be upregulated even after expression of canonical neuroectoderm marker SOX1, suggesting a unique bi-potent cell state.

## Results

### Multidimensional single cell protein data reveals a spectrum of differentiating ESCs that cannot be adequately represented by dimension reduction methods

We performed timelapse microscopy of ESCs undergoing differentiation in either retinoic acid (N2B27 basal medium + 0.1 uM retinoic acid[5], "RA"), or mesoderm (N2B27 basal medium + 100 ng/ul Activin-A + 3 uM CHIR99021[5,26], "Meso") or self-renewal promoting ("SerumLIF") medium for two days followed by IMC to obtain single-cell 37-dimensional protein quantifications of molecular markers and regulators (Fig. 1A). We used NANOGVenus/iRFPnucmem reporter ESCs[24,27] to quantify the single-cell expression dynamics of the pluripotency marker and regulator Nanog in these differentiating cells. In these ESCs, NANOGVenus enables long-term continuous quantification of live cell NANOG expression dynamics by time lapse fluorescence microscopy, nucleus segmentation via iRFPnucmem, cell tracking and quantification. Thus, for each cell, lineage information, NANOG protein levels at each timepoint and end-point multi-dimensional protein data was available to better understand the process of ESC differentiation (Fig. 1B).

A panel of 37 antibodies, now available as a resource for future studies, was optimized for IMC to detect pluripotency factors, differentiation markers, signaling molecules, histone modifiers and cell cycle markers, providing a comprehensive protein snapshot of cell

can simultaneously study only very few molecular regulators over time or many regulators at only individual time points[15–17]. While both approaches have their individual value, they lack crucial information about cellular and molecular dynamics, or about a more complete picture of molecular cell states. More recently, single cell omics approaches provide a comprehensive view of cell states[18,19], but lack information about cellular dynamics, history, kinship and of the differentiation path cells took to arrive at their current cell state. This is

state (Fig. 1B, Supplementary Table 1). Data was obtained from two biological replicates having similar protein expression profiles with no batch effects (Supplementary Fig. 1A). IMC revealed heterogeneous expression of most proteins, even in neighboring cells.

Protein expression profiles after 46 h of differentiation were highly heterogeneous between single ESCs within the same and across different media conditions (Fig. 1C, Supplementary Fig. 2A). For example, KLF4, a naïve pluripotency factor[16], was downregulated in RA and Meso compared to SerumLIF media (Fig. 1C). However, SOX2, another pluripotency factor with a role also in neuroectoderm differentiation[5], was similarly expressed in all media with only a small population downregulating it during differentiation (Fig. 1C).

Dimensionality reduction of all cells based on non-discretized IMC protein data lead to a general separation by media conditions, but visually no distinct fate enriched populations indicated by neuroectoderm (Sox1) and endoderm fate marker expression (Sox17, FoxA2) separate from the overall population were observed (Fig. 1D, Supplementary Fig. 2B). Rather, the visually distinct populations are groups of mitotically active cells indicated, for example, by upregulated phHistoneH3 expression (Fig. 1D). At the same time, there is a huge variation in range per antibody with Sox17 having range of ~0 to 0.5 while phHistoneH3 has a range of ~0–30 (Fig. 1D). This is likely due to the quality of the individual antibody, the metal isotope it is tagged with, as well as true biological variation. Prior to dimensionality reduction, we performed range scaling of antibody expression (See Methods for more details) to give equal weightage to each antibody. However, this can also artificially increase the range such that heterogeneity is observed where no true biological heterogeneity exists such as, perhaps, in the case of Sox17 and Sox1. Therefore, to overcome this problem and identify biologically relevant marker-specific cell states, we decided to binarize the antibody data. We determined each protein as non- or expressed based on individual thresholds, and classified cells according to their combinatorial thresholded protein expression state. For markers which are upregulated upon differentiation, such as SOX1, the threshold was determined based on previous knowledge and examination of raw images (Fig. 1E): Since expression is not expected in pluripotency media, thresholds were set on data from SerumLIF conditions to minimize false positives. As an example, SOX1 expression was mostly absent in both, pluripotency, and mesoderm promoting media, while retained in RA where neuroectoderm differentiation and thus SOX1 expression is expected[28]. For proteins that are highly expressed in pluripotent ESCs, expression data per condition was fitted with gamma distributions and the intersection of these fitted curves was taken as the threshold (Fig. 1F, Supplementary Fig. 3A). Preferred cell states based on this binary classification of proteins reveals a core, robust pluripotency network of NANOG, OCT4, SOX2 and ESRRB that persists in a subset of cells even after 48 h in differentiation media (Supplementary Fig. 3B, See Source Data for complete list of all cell states per media and their proportions)

## NANOG downregulation occurs two generations prior to but is not sufficient for neuroectoderm marker SOX1 upregulation

Following manual threshold assignment of end point IMC data, we performed a plausibility check to ensure our thresholds values were correctly assigned. Tracking and quantifying single-cell the expression of the pluripotency marker NANOG over 46 h in SerumLIF, we observed a bimodal population, with NANOG+ or NANOG- cell proportions remarkably consistent across generations (Fig. 2A). Owing to artefacts due to morphological changes during mitosis when the nuclear envelope breaks, NANOG reporter expression in the last two timepoints prior to mitosis and the first two timepoints after mitosis were always excluded for any tracked cell. The proportions obtained with NANOG reporter were consistent with those obtained by end point NANOG IMC data (Supplementary Fig. 3C). Two independent

measurements and two independent thresholds yielding the same results validated our strategy for threshold assignment.

The evidence of a persistent NANOG- population in pluripotent media is in line with previously described varied NANOG dynamics[24]. Utilizing NANOG reporter expression and lineage data, we computed NANOG-based state transition probabilities across generations in SerumLIF pluripotency media (see Methods for details). The probability of NANOG+ cell transitioning to NANOG- cell in the next generation is very low ($0.05 \pm 0.01$) compared to a NANOG- cell transitioning to NANOG+ cell ($0.3 \pm 0.07$). This is not unexpected as, even if Nanog is downregulated in one generation, the surrounding network of pluripotent proteins and the media conditions themselves, likely keep the cell in a NANOG+ state which is a more stable state in these culture conditions (Supplementary Fig. 3B). The presence of a minor NANOG- negative population in SerumLIF, that doesn't increase in proportion with increasing generations (Fig. 2A, Supplementary Fig. 3C), is also indicative of the unstable nature of a NANOG- state in pluripotent media. Further supporting our threshold strategy, this is corroborated by inferring NANOG transition dynamics following Kin Correlation Analysis (KCA)[29] using end point NANOG IMC data and lineage information. Here, the inferred probability of a NANOG+ cell transitioning to NANOG- cell is $0.03 \pm 0.01$ and for a NANOG- cell transitioning to NANOG+ cell it is $0.24 \pm 0.07$.

Next, we analyzed NANOG dynamics also in ESCs cultured differentiation media RA. As expected, and different from cultures in pluripotency maintaining media, cells mostly downregulated NANOG in RA, with only a minor persisting NANOG+ population (Fig. 2B, Supplementary Fig. 3D). NANOG expression thresholds were set as described above, and IMC-based thresholding again confirmed the threshold based on live cell NANOG quantification with an accuracy of 85.2%. Calculating NANOG-based state transition probabilities as above for RA, we found that, opposite to transition probabilities in SerumLIF, in RA differentiation media, the probability of NANOG+ cell transitioning to NANOG- cell in the next generation is much higher ($0.26 \pm 0.09$) compared to a NANOG- cell transitioning to NANOG+ cell ($0.1 \pm 0.07$). In Meso differentiation medium, the transition probabilities are roughly similar, with NANOG- to NANOG+ being $0.16 \pm 0.04$ and NANOG- to NANOG+ being $0.12 \pm 0.05$. This is likely because of the presence of CHIR99021 inhibitor in Meso differentiation medium, which is also a key component of the pluripotency promoting defined media 2iLIF[30].

We next investigated NANOG dynamics in predicting SOX1 neuroectoderm marker expression in RA medium. Based on kinship information from tracking data (46 h in RA), we observed that usually only some related cells in colony sub-branches were SOX1+. In these colony sub-branches, the decision for SOX1 upregulation most likely occurred in a common ancestor of these SOX1+ cells rather than independently in every related cell. We termed these common ancestors of SOX1+ progeny 'Decision Cells' (Fig. 2C). NANOG reporter expression revealed that nearly all Decision Cells are NANOG- (97%) (Fig. 2C). When including data from the previous generation, 95% of Decision Cells and their parents are NANOG- (Fig. 2C). When including data from two previous generations, the percentage of Decision Cells and their ancestor cells that are all NANOG- is still 87% (Fig. 2C). This reduces to 69% and 50% when including data from three and four previous generations, respectively (Fig. 2C). Thus, NANOG downregulation has typically already occurred two generations prior to SOX1 upregulation. We then asked if NANOG downregulation for two generations is sufficient for SOX1+ commitment by quantifying NANOG dynamics of all SOX1- cells and their ancestors (Fig. 2C). While not as pronounced as with Decision Cells, a majority of SOX1- cells are also NANOG- (87%) (Fig. 2C). 82% of SOX1- cells and their parents are NANOG-. These observed proportions for SOX1- cells and their ancestors one generation up are significantly different compared to SOX1+ Decision Cells but the effect size is minor (10% difference at

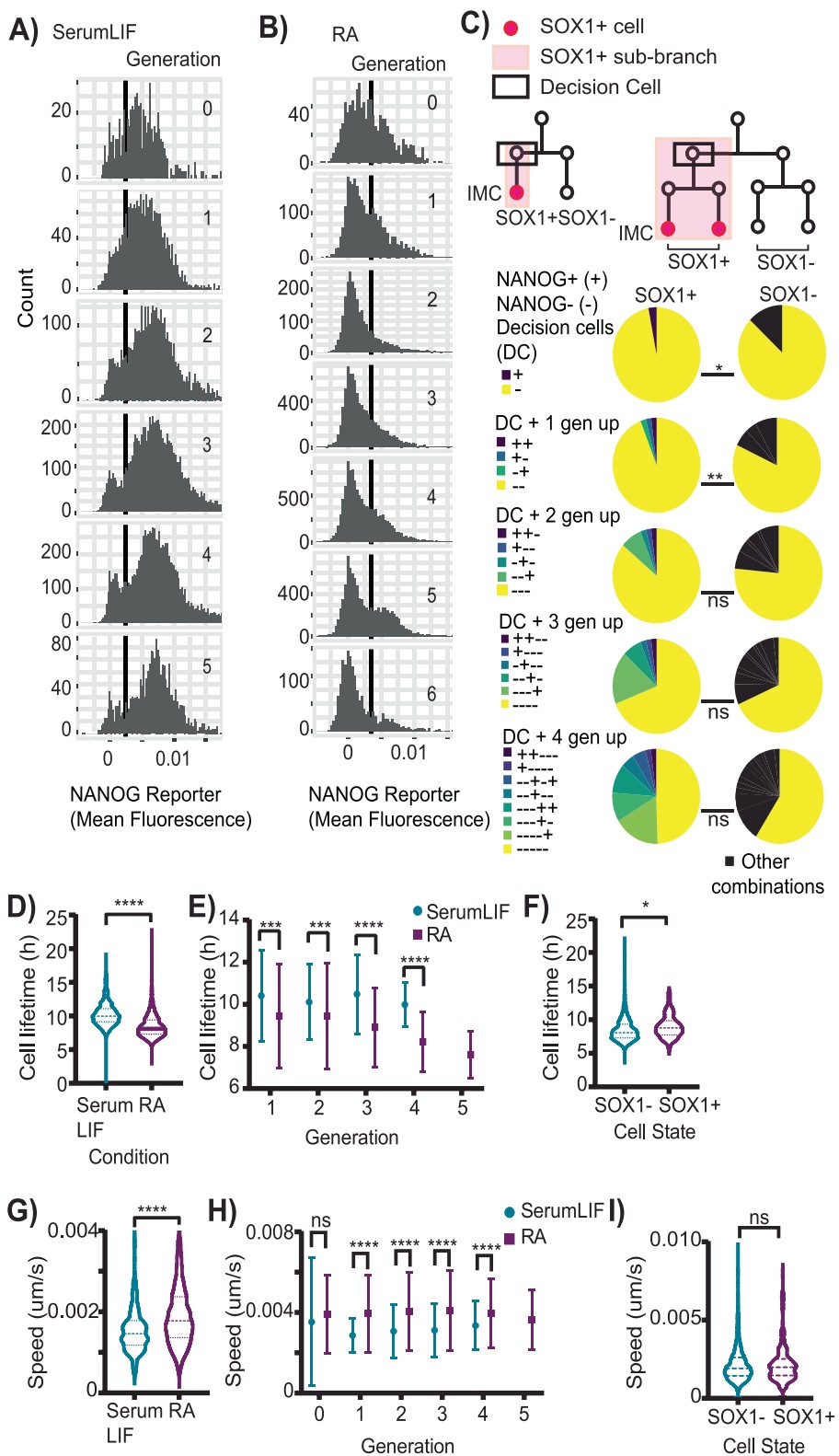

current generation and 13% difference at current generation plus parent) and doesn't have predictive power. The difference in proportions becomes non-significant when considering ancestral dynamics from 2 generations up and higher. 72% are NANOG- for two previous generations, 69% for three previous generations and 59% for four previous generations (Fig. 2C). Thus, while more cells are NANOG- in the previous two generations of SOX1+ (87%) versus SOX1- cells (72%),

the observed proportions are not significantly different and NANOG- is the dominant state in both cases (Fig. 2C). Thus, NANOG down-regulation two generations prior is not sufficient for starting SOX1+ expression.

We next looked at cellular dynamics such as cell lifetime and cell motility to check if they can predict neuroectoderm marker SOX1+ expression. All cells of SOX1+ Decision Cell sub-branches were labelled

**Fig. 2 | Nanog dynamics, cell motility and lifetime are not indicators of neuroectoderm marker Sox1 expression. A, B** Live cell NANOG reporter expression per generation. Manual threshold (black). N > 25000/ > 40000 data points from complete lifetimes of each tracked cell in 52/71 trees from 2/2 biological replicates in A (SerumLIF)/B (RA). **C** NANOG downregulation occurs two generations prior to, but is not sufficient for, Sox1 expression. Ancestral NANOG levels of SOX1+ Decision Cells (left) and SOX1- cells (right). Decision Cells (rectangle box) detected as shown graphically (see Methods). "+-" means current cell is NANOG+ and its mother cell is NANOG-, and so on. N = 66 Decision Cells and 294 SOX1- cells from two biological replicates. *p = 0.04, **p = 0.007, ns: not significant p = 0.06, 0.87, 0.31 (2, 3, 4 gen up), Two-sided Fisher's exact test, with cells in yellow as category 1 and all other combinations grouped together. **D** ESCs in SerumLIF have longer cell lifetimes. N > 850 cells per condition from 2 biological replicates. Dotted lines represent 25th, 50th and 75th percentile values. ****p < 0.0001, Two-tailed unpaired t-test. **E** Difference in cell lifetime persists over generations. From generation 1–4, n = 102, 200, 325, 233 cells in SerumLIF and from generation 1–5, n = 136, 246, 398, 517, 259 cells in RA from 2 biological replicates. Error bars: means ± SDs. ****p < 0.0001, ***p = 0.0002, 0.0005 (Gen 1, 2), 2-way ANOVA corrected for multiple comparisons. **F** SOX1+ and SOX1- cells in RA have similar lifetimes. N = 120 SOX1+ and n > 1500 SOX1- cells from 2 biological replicates. Dotted lines: 25th, 50th and 75th percentile values. *p = 0.02, Two-tailed unpaired t-test. **G** Cells in RA are more motile. N > 900 cells per condition from 2 biological replicates. Dotted lines: 25th, 50th and 75th percentile values. ****p < 0.0001, Two-tailed unpaired t-test. Graph magnified. Original in Supplementary Fig. 3E. **H** Difference in cell motility in persists over generations. From generation 0-4, n = 52, 102, 200, 325, 233 cells in SerumLIF and from generation 0-5, n = 70, 136, 246, 398, 517, 259 cells in RA from 2 biological replicates. Error bars: means ± SDs. ****p = 6.76E-06, 6.45E-09, 1.23E-13, 4.55E-05 (Gen1, 2, 3, 4), ns: not significant p = 0.71, 2-way ANOVA corrected for multiple comparisons. **I** SOX1+ and SOX1- cells in RA have similar speeds. n > 300 SOX1+ and n > 3000 SOX1- cells from 2 biological replicates Dotted lines represent 25th, 50th and 75th percentile values. ns: not significant p = 0.95, Two-tailed unpaired t-test.

SOX1+. The remaining cells were classified SOX1-. For cell lifetime (time between two mitosis events) analysis, only dividing cells were included whose parent could be identified. Cells in RA had a significantly shorter lifetime (8.6 ± 1.9 h) than in pluripotency media (10 ± 1.8 h) (Fig. 2D), as previously reported[31]. This difference is present across generations and increases with increasing generation numbers (Fig. 2E). However, differences between cell lifetimes of SOX1+ (8.9 ± 1.6 h) and SOX1- cells (8.5 ± 1.9 h) (Fig. 2F) within the same RA condition were minor (24 minutes). Cell cycle lengths thus do not identify this neuroectoderm marker expression. Similarly, cell motility, which is related to cell adhesion and is defined as mean distance travelled per timelapse imaging frame, is significantly higher in RA (0.002 ± 0.0006 um/s) than pluripotency media (0.0013 ± 0.0006 um/sec) (Fig. 2G), and this is consistent across generations (Fig. 2H). However, cell motility is not different between SOX1+ and SOX1- cells in RA (Fig. 2I), and thus not useful as an indicator of SOX1+ upregulation.

## Unexpected co-expression of neuroectoderm (SOX1) and endoderm (FOXA2) markers in rare differentiating ESCs

In our analysis of Decision Cells in the previous section, we identified 28 pairs of sister cells with differential SOX1 expression in the last generation for which IMC data is available (Fig. 3A). This enables the identification of individual cells in a specific interesting time window: since these sister cells were the same single (mother) cell, their asymmetric SOX1 expression indicates that the fate decision leading to SOX1 expression happened in only one of the sisters and only in the few hours since division. The sister cells should thus differ mostly in their molecular programs associated with that state change, potentially allowing the identification of relevant regulators or co-regulated molecules[23]. However, we observed no significant difference for any of the analyzed pluripotency factors or signaling molecule (Fig. 3A). We found significant differential expression for cell cycle marker Ki67[32] and histone modifier Dnmt3b[33] (Fig. 3A). However, unlike for SOX1, the overall distribution of Ki67 and Dnmt3b is relatively similar between the Sox1+ and Sox1- sister pairs (Fig. 3A). The significance therefore likely arose from a few outliers affecting the mean in this small population.

We also quantified the time since division of the SOX1+ cells in these sister pairs to determine if there was a particular time window during which these cells committed to SOX1+ upregulation, indicative of neuroectoderm lineage (Fig. 3B). Time since division of these cells in the last generation prior to IMC were spread from 30 minutes to 10 hours (Fig. 3B) indicating cell cycle stage independent cell SOX1 upregulation. Asymmetric fate choices between two sisters very early in their cell cycle demonstrates rapid fate decision making in the very short time since their mother's division.

In addition to SOX1 expressing cells, which is expected in neuroectoderm promoting RA, we also observed a considerable proportion of cells (4%) expressing the endoderm marker FOXA2 (Fig. 3C)[34]. Extra-embryonic endoderm cells (XEN), resembling primitive endoderm, are of interest in the field and share markers with definitive endoderm lineage (SOX17)[35]. They are also known to arise from RA media, though at a later time of 4 days[36]. We therefore checked if FOXA2 was coexpressed with the XEN markers SOX17 and GATA6, but this was not the case (Fig. 3C). In addition, we repeated the analysis performed earlier with SOX1+ cells (Figs. 2C, F, I and Fig. 3A), to identify if NANOG dynamics, lifetime, motility or cell marker can predict expression of endoderm marker FOXA2, mesoderm marker BRACHYURY or any germ layer marker in general (SOX1+/FOXA2/BRACHYURY+/GATA6+/SOX17+ ) (Supplementary Fig. 4A-K). Our analysis yielded similar results as with SOX1 and no predictive indicators of the germ layer markers in the IMC panel were found.

Our IMC quantification revealed a small population of cells (1%) to be co-expressing both SOX1 and FOXA2 markers (Fig. 3C). This co-expression was confirmed by manual inspection of raw images, and proteins localized to nucleus as expected (Fig. 3C). For meso-endoderm differentiation, co-expression of meso- and endoderm markers has been described in an intermediate population later giving rise to committed endo- or mesodermal cells[37]. Therefore, the rare cells co-expressing SOX1 and FOXA2 could be precursors to uni-marker cells expressing either only neuroectoderm marker SOX1 or only endoderm marker FOXA2. SOX1+FOXA2+ cells were always present as sister to SOX1+, FOXA2+ or SOX1+FOXA2+ cells, but never in pairs with SOX1-FOXA2- cells. This suggests that they may be preceding uni-marker expression between ectoderm marker SOX1 and endoderm marker FOXA2 (Fig. 3D). To rule out that this may be an artefact of the IMC protocol or an occurrence in this reporter line only, we performed immunostaining for SOX1, FOXA2 and NANOG of R1WT ESCs (parental line of R1 NANOGVenus/iRFPnuc-mem) in RA media (Fig. 3E) and NG4 cells which come from a different strain[38]. In all cases, we detected FOXA2+ and SOX1+FOXA2+ co-expression (Fig. 3E).

We repeated ESC cultures with different concentrations of RA to check if increased RA correlates with increases in any progenitor cell population. Positive NANOG expression was scored based on our positive control of SerumLIF (Fig. 3F). NANOG- cells were then assigned SOX1 and FOXA2 threshold values based on the negative control of N2B27 media (the basal media of RA) and the proportion of SOX1+ , FOXA2+ or SOX1+FOXA2+ cells were quantified (Fig. 3F). Increasing RA from 0.1 to 0.5 uM led to an overall increase in cells expressing differentiation markers (Fig. 3G). This difference was less pronounced upon increasing RA concentration to 1 uM (Fig. 3G). Nevertheless, for all RA concentrations and two different cell lines, the SOX1+FOXA2+ population was still observed, confirming that our initial observation from IMC data was not an artefact but potentially identified a rare developmental cell differentiation stage.

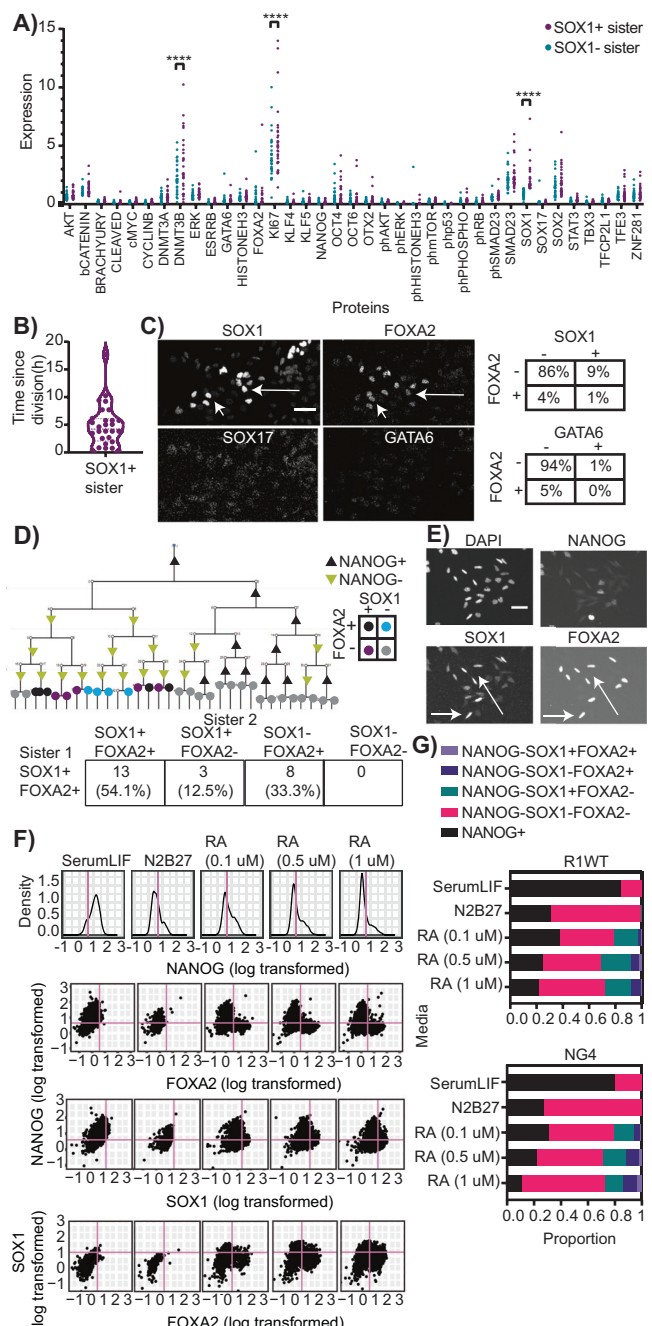

**Fig. 3 | A cell population co-expressing canonical ectoderm and endoderm lineage markers. A** No significant differences in pluripotency marker and signaling molecule expression of SOX1+ /SOX1- sister pairs. n = 28 sister pairs from two biological replicates. ****p < 0.0001 where indicated, else non-significant p > 0.99, 2-way ANOVA corrected for multiple corrections. **B** Neuroectoderm marker Sox1 does not get upregulated at a specific cell cycle stage. **C** A significant proportion of cells in neuroectoderm promoting RA media express the FOXA2 endoderm marker. Rare cells co-express SOX1 and FOXA2 (arrows). FOXA2 is not co-expressed with endoderm marker SOX17 or primitive endoderm marker GATA6 (shown dim signals for both are only background). n > 4000 NANOG- cells from 2 biological replicates. Scale bar 50 um. **D** Observed SOX1+FOXA2+ cells are closely related to single positive or other double positive cells, but never to SOX1-FOXA2- cells, indicating a potential progenitor population of uni-marker expressing cells. Phenotype frequencies of sisters of SOX1+FOXA2+ cells (bottom). n = 24 sister pairs from 2 biological replicates. **E** The parental line R1WT also differentiates into SOX1+FOXA2+ (arrows), -+ and +- cells. Representative immunostaining images after 48 h in RA, n = 2 biological replicates. Scale bar 50 um. **F** Cell state classification based on NANOG, FOXA2 AND SOX1 immunostaining quantification after 48 h in different media conditions. NANOG threshold (pink) assigned based on NANOG expression in SerumLIF (top) and FOXA2 (second from top) and SOX1 (second from bottom) threshold based on their expression in N2B27 alone. Same thresholds used for SOX1+FOXA2+ population in RA media (bottom). n > 700 cells per condition from one biological replicate. **G** SOX1+FOXA2+ cells are generated in different RA concentrations. Proportion of cells in R1WT (top) and NG4 (bottom) ESC lines. Thresholds as shown in F) and G). n > 700 cells per condition per cell line from 1 biological replicate each. A 2nd replicate yielded similar numbers (Supplementary Fig. 4L).

reporter marker expression and then performed antibody staining against the proteins of interest (Fig. 4B). FOXA2mCherry was correctly localized in the nucleus with a high degree of correlation between reporter (absent in the parental line as expected) and antibody (r = 0.84) (Fig. 4B, C). Since EGFP is not fused to SOX1 protein itself, it locates to the whole cell body as expected. Its expression correlated with endogenous SOX1 protein expression as detected by antibody and was similar to the parental line (Fig. 4B, C). Again, we saw the presence of SOX1+FOXA2+ cells also in this double reporter line (Fig. 4C). We repeated this quantification at 4 and 6 days (D4/D6) post RA differentiation to exclude that the fluorescence reporter intensity gets decoupled from protein expression over time (Supplementary Fig. 5A). FOXA2mCherry expression remained well correlated with FoxA2 protein levels at both D4 (r = 0.7) and D6 (r = 0.6). Based on examining the raw images, we attribute the gradual decrease in r values over time to increased cell density, cell debris autofluorescence and more diverse morphologies over time, making image segmentation and quantification noisier. Sox1-EGFP expression remains well correlated to Sox1 protein expression at D4 (r = 0.6) but not at D6 (r = 0.29) (Supplementary Fig. 5A). As EGFP is not fused to Sox1 itself, its expression potentially gets decoupled from SOX1 protein expression at D6. Performing time-series FACS analysis to identify the proportion of Sox1 + FOXA2-, Sox1 + FOXA2- and Sox1 + FOXA2+ cells over time (Supplementary Fig. 5C, D), we observed that the percentage of Sox1+FOXA2- cells peaks at 2 days post differentiation before gradually returning to baseline by D6 (Supplementary Fig. 5C), consistent with literature[40]. In contrast, the proportion of Sox1-FOXA2+ cells increases with time (Supplementary Fig. 5C). These observed frequencies might also explain the low correlation of Sox1-EGFP to SOX1 protein expression at D6. As the overall expression levels of EGFP and SOX1 antibody are at their lowest at D6 (Supplementary Fig. 5A) with many cells Sox1-, quantification becomes noisier as the signal is close to background.

To check the timing of Sox1 and FOXA2 upregulation, we tracked individual cells throughout differentiation and quantified their fluorescent expression. We observed both, Sox1 upregulation without prior FOXA2 expression and FOXA2 upregulation without prior Sox1 expression (Fig. 4D), thus ruling out SOX1+FOXA2+ cells as mandatory

## SOX1 expressing cells can still develop into FOXA2 positive cell types

Our data suggested that SOX1+FOXA2+ co-expression might potentially identify a bipotent precursor stage with ecto- and endodermal differentiation potential marked by SOX1 and FOXA2 expression. However, better understanding these cells' possible bipotentiality requires continued live single cell quantification of these transcription factor expression dynamics. We therefore engineered a Sox1-EGFP/FOXA2mCherry double reporter line. We used the previously published Sox1-EGFP line where GFP is knocked in downstream of the Sox1 promoter in its endogenous locus, replacing the Sox1 gene in one endogenous allele[39]. Using Crispr-Cas9-mediated homologous recombination, we fused mCherry to the C-terminus of FOXA2 by knockin into the FoxA2 locus (Fig. 4A). To ensure FOXA2mCherry accurately reports FOXA2 protein expression with nuclear localization and expression onset as previously observed in RA medium, we subjected cells to RA differentiation for 2 days, tracked fluorescent

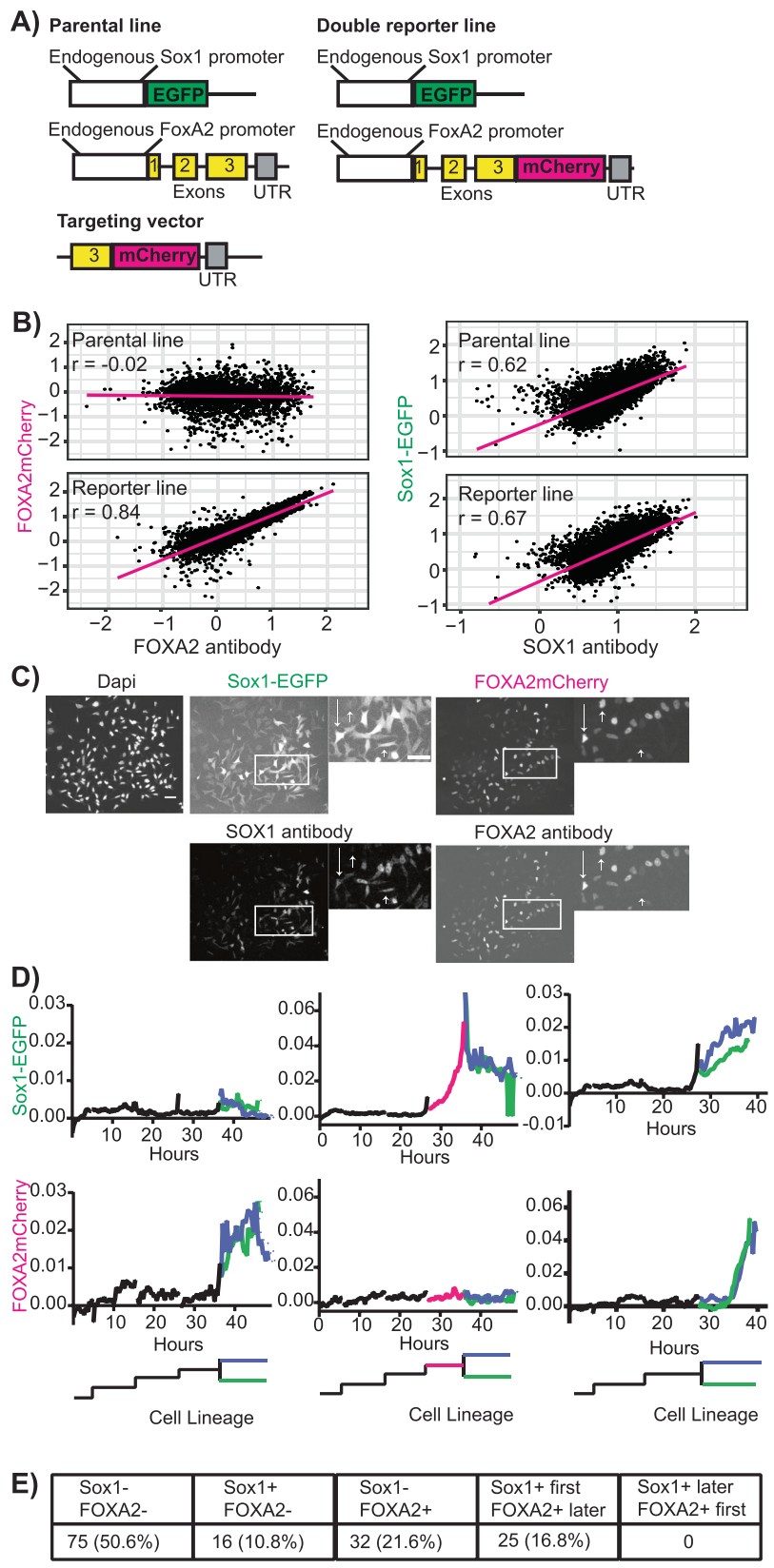

| Sox1- FOXA2- | Sox1+ FOXA2- | Sox1- FOXA2+ | Sox1+ first FOXA2+ later | Sox1+ later FOXA2+ first |
|---|---|---|---|---|
| 75 (50.6%) | 16 (10.8%) | 32 (21.6%) | 25 (16.8%) | 0 |

differentiation stage preceding ecto- versus endodermal uni-marker expression. In addition, we observed FOXA2 upregulation following Sox1 expression in the same cells in 16.8% of all cell tracks analyzed (Fig. 4E). Interestingly, the opposite occurrence – FOXA2 expression before Sox1 upregulation – was never observed for any of the tracked Sox1+ cells (Fig. 4E). This suggests that FOXA2 expression potentially reports the loss of differentiation potential into neuroectoderm marker Sox1 positive cells, while Sox1+ expressing cells can still differentiate into the endoderm marker FOXA2 positive state. This cautions us from using SOX1 alone as a marker for neuroectoderm cells and indicates that there are multiple or more flexible routes to endodermal differentiation.

**Fig. 4 | Sox1 expressing cells can still differentiate into a FOXA2 positive endodermal marker state. A** Knock-in targeting strategy to obtain a Sox1-EGFP/FOXA2mCherry double reporter line using Crispr-Cas9. **B** The Sox-EGFP/FoxA2mCherry line accurately reports FOXA2 and SOX1 protein expression. Expression quantified 2 days post RA differentiation. Log transformed data plotted. Linear regression line (pink) and Pearson correlation coefficient (r) as indicated. n > 2500 cells per line from two biological replicates. **C** Representative images of data quantified in B) showing SOX1 and FOXA2 reporter expression as well as SOX1 and FOXA2 immunostaining data for the same cells. Inset shows images at higher magnification. Scale bars 50 um. Overlay images in Supplementary Fig. 5B. **D** Quantification of Sox1-EGFP (top) and FOXA2mCherry (bottom) expression dynamics in single cells. Timelapse imaging and cell tracking during RA differentiation. Only partial cell pedigrees shown for clarity. Representative examples show upregulation of FOXA2 alone (left), SOX1 alone (middle) and both SOX1 and FOXA2 (right). **E** SOX1+ cells upregulate FOXA2 but not vice versa. n = 148 cells from two biological replicates.

## RNASeq reveals that SOX1-FOXA2+ cells have visceral/parietal endoderm lineage potential while SOX1+FOXA2+ cells are characterized by expression of both endoderm and neuroectoderm markers

To get a more comprehensive characterization of the cell state of these Sox1+FOXA2-, Sox1-FOXA2+ and SOX1+FOXA2+ cells, we differentiated Sox1-EGFP/FoxA2mCherry cells for 2, 4 or 6 days in RA medium (D2/D4/D6), FACS sorted for desired populations and performed bulk RNASeq (Fig. 5A). In addition, to characterize the lineage potential of these cells, we differentiated the reporter line for 2 days in RA, sorted for desired populations, and cultured them in RA medium for a further 2 or 4 days for a total differentiation time of 4 or 6 days (Progeny D4/D6), followed by RNASeq (Fig. 5A). PCA analysis of RNASeq samples revealed that all replicates per sample generally cluster together based on Principal Components (PCs) 1 and 2 which explain 38% and 18% of the total variance observed in all the samples respectively (Fig. 5B). Examining the trend of how the samples lie on PC1 and PC2, it is apparent that PC1 explains changes in cell populations over time as samples move from left to right from D0 to D6 (Fig. 5B). PC2 clearly distinguished the Sox1+FOXA2- populations from Sox1-FOXA2+ populations. These samples lie close to each other at D2 and in the middle of PC2 but start to diverge with Sox1-FOXA2+ samples moving up PC2 with time while Sox1+FOXA2- samples move down PC2 (Fig. 5B). The Sox1-FOXA2- samples lie on the same trajectory as the Sox1+FOXA2- cells which is to be expected as the medium is primarily neuroectoderm promoting (Fig. 5B). Sox1+FOXA2+ cells lie closer to Sox1+FOXA2- cells in PC space, but indeed lie in between Sox1-FOXA2+ and Sox1+FOXA2- samples, with this population being the most variable at D6 (Fig. 5B).

A similar trend is also observed in the progeny of D2 sorted populations at D4 and D6 with the progeny of Sox1+FOXA2+ cells in between the progeny of the other two sorted populations, particularly apparent in D6 progeny (Fig. 5B). Examining PC2, which clearly separates the sorted populations, we observed that the top positive loadings of PC2 include many endoderm markers including Gata4, Lamb1, Dab2, Sox7, Fst, Gata6 as well as FoxA2, while the top negative loadings of PC2 include many genes associated with neural populations such as Nestin (Nes), Meis2, Sox9, Cpe and Nr2f2. Thus, populations sorted on the basis of the two markers Sox1 and FOXA2 reflect broad transcriptomic changes with the progeny of the sorted population also adopting distinct fates.

Examining the individual expression levels of key marker genes also confirmed the dynamics of gene expression observed from PCA analysis (Fig. 5C, D, Supplementary Fig. 6A, B). Expression of key markers for endoderm and neuroectoderm is not visually distinct between the Sox1+FOXA2-, Sox1-FOXA2+ and Sox1+FOXA2+ cells at D2. But by D4 and D6, clear upregulation of endoderm markers is observed in Sox1-FOXA2+ cells, while neuroectoderm markers are upregulated in Sox1+FOXA2- cells. Sox1+FOXA2+ cells display a unique cell state wherein both neuroectoderm and endoderm markers are expressed at intermediate levels (Fig. 5C, D, Supplementary Fig. 6A, B). As expected, sorted FOXA2mCherry+ populations show higher levels of FoxA2 mRNA transcripts across time than FOXA2- populations, further validating the FOXA2mCherry reporter (Fig. 5D). We did not detect Sox1 RNA in our dataset. We have shown earlier that it is expressed at protein level (Fig. 4C) and well correlated to EGFP

reporter fluorescence levels. In addition, other neuroectoderm markers like Nestin and Pax6, a key gene in neurogenesis expressed sequentially after Sox1[41], are upregulated in populations sorted for Sox1-EGFP+. Since EGFP replaces one Sox1 allele in the reporter line, it is possible that Sox1 mRNA levels were below RNASeq detection levels. Interestingly, IMC data at D2 showed FOXA2 expression without co-expression of other endoderm markers such as GATA6 and SOX17 (Fig. 3E). Examining the RNA data, we see that while FoxA2 mRNA levels remain steady across time in Sox1-FOXA2+ cells, Sox17 and Gata6 mRNAs are present in low amounts at D2 and increase gradually with time (Fig. 5D). This suggests that in endoderm lineage commitment in RA, decisions are initiated by FoxA2 while other endoderm markers are expressed later with lineage maturation. We did not detect expression of meso-endoderm markers such as Brachyury, Gsc and Eomes[37,42], or markers for other populations that express FoxA2 such as node, notochord etc, which all originate from the mesoderm[43]. Instead, we observe a strong upregulation of many markers expressed in visceral and parietal endoderm (VE and PE). We therefore conclude that the Sox1-FOXA2+ populations are VE/PE.

Next, we checked if Sox1 and FOXA2 expressing cells at D2 are already committed to giving rise to neuroectoderm and endoderm cells, even though, as characterized by entire transcriptome, the sorted populations lie close to each other in PCA space, and many lineage markers have not yet segregated. D4 and D6 transcriptomes of the progeny of D2 Sox1+FOXA2- and Sox1-FOXA2+ cells revealed homogeneous lineage signatures within their group, but distinct between these groups. The same was true for their morphologies (Fig. 5E–G, Supplementary Fig. 6D–F). Sox1+FOXA2- cell progeny have a flat, elongated morphology (Fig. 5G, Supplementary Fig. 6F), typical of neural cells[44], and show upregulation of neuroectoderm markers (Fig. 5E, F, Supplementary Fig. 6D, E). In contrast, Sox1-FOXA2+ cell progeny has compact, epithelial-like morphology (Fig. 5G, Supplementary Fig. 6F), typical of endoderm cells[45], and show upregulation of endoderm markers (Fig. 5E, F, Supplementary Fig. 6D, E). Sox1+FOXA2+ cell progeny have a largely neuroectodermal signature, but with expression of endoderm markers higher than Sox1+FOXA2- progeny and lower than Sox1-FOXA2+ progeny (Fig. 5E, F, Supplementary Fig. 6D, E). Most of the Sox1+FOXA2+ progeny have a morphology consistent with that observed in Sox1+FOXA2- progeny, but we also observe a small subset of cells to be compact and epithelial-like, as seen in Sox1-FOXA2+ cells (Fig. 5G, Supplementary Fig. 6F). This suggests that Sox1+FOXA2+ cells reflect a unique bi-potent cell state.

## SOX1+FOXA2+ cells exist as a specific cell type also in vivo

The dynamics of Sox1 and FOXA2 expression and the transcriptomes of cells expressing these markers in cultures of differentiating mESCs suggested that SOX1+FOXA2+ coexpression marks a specific developmental cell state or type. We therefore wanted to determine if SOX1+FOXA2+ cells are also present in embryos, and at which specific times and locations. To this end, we stained whole mouse embryos for SOX1 and FOXA2 and imaged every cell in the embryo at different stages from embryonic day (E)6.5 to E7.5 as well as neural tube slices at E8.5 (Fig. 6, Supplementary Fig. 7, Supplementary Movie 1). At E6.5 – E8.5, embryos have implanted in the uterus, gastrulation commences initiated by primitive streak formation, and pluripotent cells

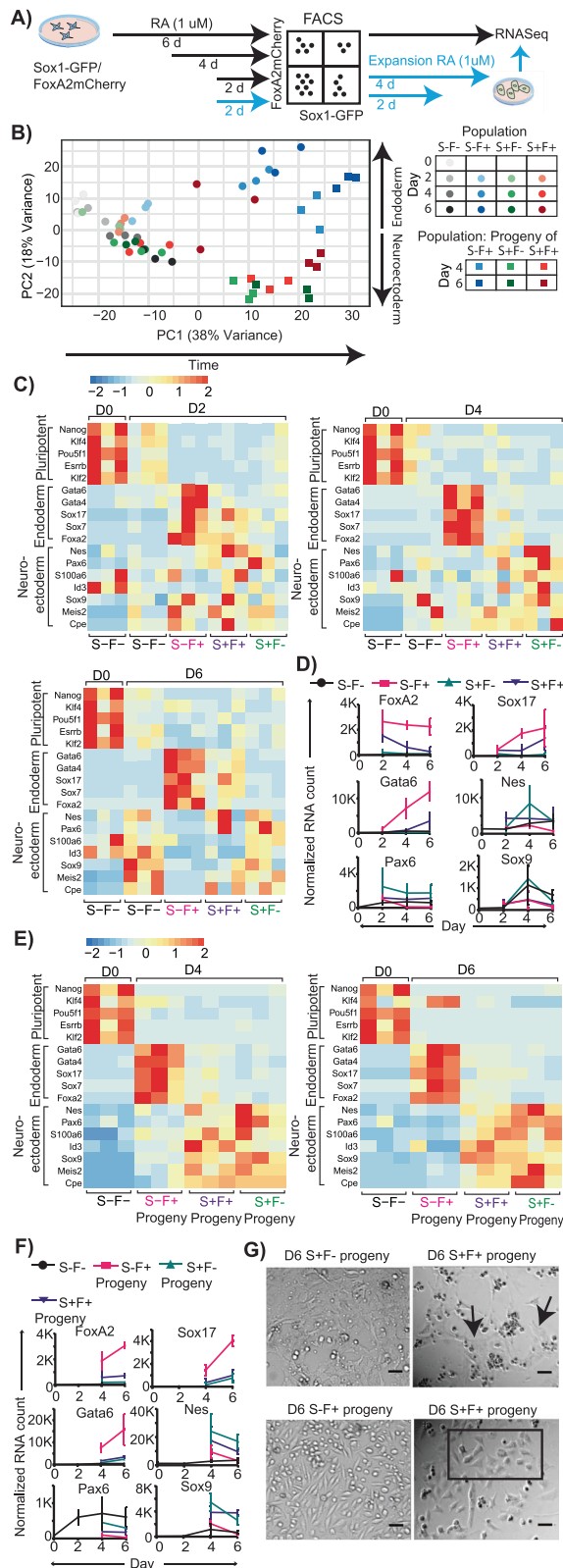

**Fig. 5 | Sox1+FOXA2+ cells express both neuroectodermal and endodermal markers at intermediate levels compared to Sox1+FoxA2- and Sox1-FOXA2+ cells. A** Experimental workflow. Cells were either differentiated for 2/4/6 days in RA medium prior to sorting and bulk RNASeq. Or differentiated for 2 days, sorted and cultured in RA medium for another 2/4 days before RNASeq. **B** Populations sorted by marker expression cluster together in transcriptome PCA analysis, with increasing distance as differentiation time increases. Sox1+FOXA2+ cells and its progeny occupy an intermediate position in PCA space with respect to corresponding Sox1-FOXA2+ and Sox1+FOXA2- cells. n = 3 biological replicates per condition. B–G S: Sox1, F: FoxA2. **C, D** Sox1-FOXA2+ versus Sox1+FOXA2- cells exhibit a progressively stronger visceral/parietal endoderm versus neuroectoderm gene signature with increased differentiation time. Sox1+FoxA2+ cells exhibit intermediate expression for key markers of both endoderm and neuroectoderm. **C** Relative gene expression per population as indicated, following z-score transformation of normalized counts by row. **D** Unscaled gene counts, normalized only to library size. n = 3 biological replicates per condition. Error bars represent means ± SDs. **E, F** Progeny of Sox1-FOXA2+ versus Sox1+FOXA2- acquire a progressively stronger visceral/parietal endoderm versus neuroectodermal gene signature with increased differentiation time. In contrast, Sox1+FoxA2+ progeny exhibit intermediate expression for key markers of endoderm and neuroectoderm. **E** Relative gene expression per population as indicated, following z-score transformation of normalized counts by row. **F** Unscaled RNA counts over time per population. n = 3 biological replicates per condition. Error bars represent means ± SDs. **G** Sox1+FOXA2+ cells give rise to cells with two distinct morphologies: flat, elongated characteristic of neuronal cells (arrows, D6 S+F- progeny), versus epithelial-like morphology (rectangle, D6 S-F+ progeny) characteristic of endoderm cells. N = 2 biological replicates per condition. Scale bar 50 um.

expected[48]. At E7.5, FOXA2 expression was detected in the node and visceral endoderm as at E7.0 (Fig. 6A, Supplementary Fig. 7A).

We also found SOX1+FOXA2+ cells in a very specific location and developmental time window. At E7.5, a sub-fraction of the SOX1+ cells of the ventral neural groove were found to be also positive for FOXA2, with these double-positive cells being detected ventrally along the whole length of the neural groove (Fig. 6B, Supplementary Fig. 7A, Supplementary Movie 1). Scanning whole embryos, this is the only location at which SOX1+FOXA2+ cells were found.

By E8.5, the neural groove encloses dorsally to form the neural tube, with cells of the neural tube emerging into distinct neural progenitor populations[49]. p3, the ventral most progenitor population of the neural tube, is marked by Nkx2.2 expression[49]. The adjacent population, pMN, is marked by Olig2 while the more dorsal regions are marked by Pax6 expression[49]. The notochord, derived from the node, also forms as a rod like structure underlying the floor of the neural tube and gives rise to the neural floor plate[43]. Both floor plate and the notochord are marked by FoxA2 expression[43,49]. While the exact timing of how the floor plate and progenitor populations of the neural tube emerge is unclear, it is thought to follow notochord formation which secretes the morphogen Sonic Hedgehog (shh) to create a ventral-dorsal signaling gradient[49]. Given the specific ventral location of SOX1+FOXA2+ cells in the neural groove at E7.5, which precedes notochord formation, we hypothesized that these double positive cells could be p3 neural progenitors. We therefore stained E7.5 and E8.5 embryos against SOX1 and FOXA2, and NKX2.2, the marker identifying p3 neural progenitors. While no NKX2.2 expression was detected at E7.5 (Supplementary Fig. 7C), NKX2.2 was detected at E8.5 in cells at the ventral region of the neural tube (Fig. 6C, Supplementary Fig. 7B). Most of these cells were also SOX1+FOXA2+ (Fig. 6C, Supplementary Fig. 7B). The larger fraction of NKX2.2+ cells are also SOX1+FOXA2+ (70 ± 21%, n = 11 fields of view), while ~45% of SOX1+FOXA2+ cells are NKX2.2+ (45.4 ± 14.4%, n = 11 fields of view). Further splitting these embryos into younger and older around E8.5 (based on embryo size), we observed that in the younger embryos nearly all NKX2.2+ cells are SOX1+FOXA2+ (96.4 ± 7.1%, n = 4 fields of view), while in older embryos the degree of overlap reduces (55.7 ± 20.22%, n = 7 fields of view). The percentage of SOX1+FOXA2+ cells that are NKX2.2+ remains constant

differentiate into different germ layers. FOXA2 was detected at E6.5-E7.0 in the primitive streak, node, and visceral endoderm (Supplementary Fig. 7C) as expected[46,47]. No SOX1 expression was detected at these stages, indicating the neuroectoderm has not formed yet (Supplementary Fig. 7C), consistent with literature[48]. By E7.5 (early headfold stage) widespread SOX1 expression was detected in the neural groove and neuroectoderm (Fig. 6A, Supplementary Fig. 7A) as

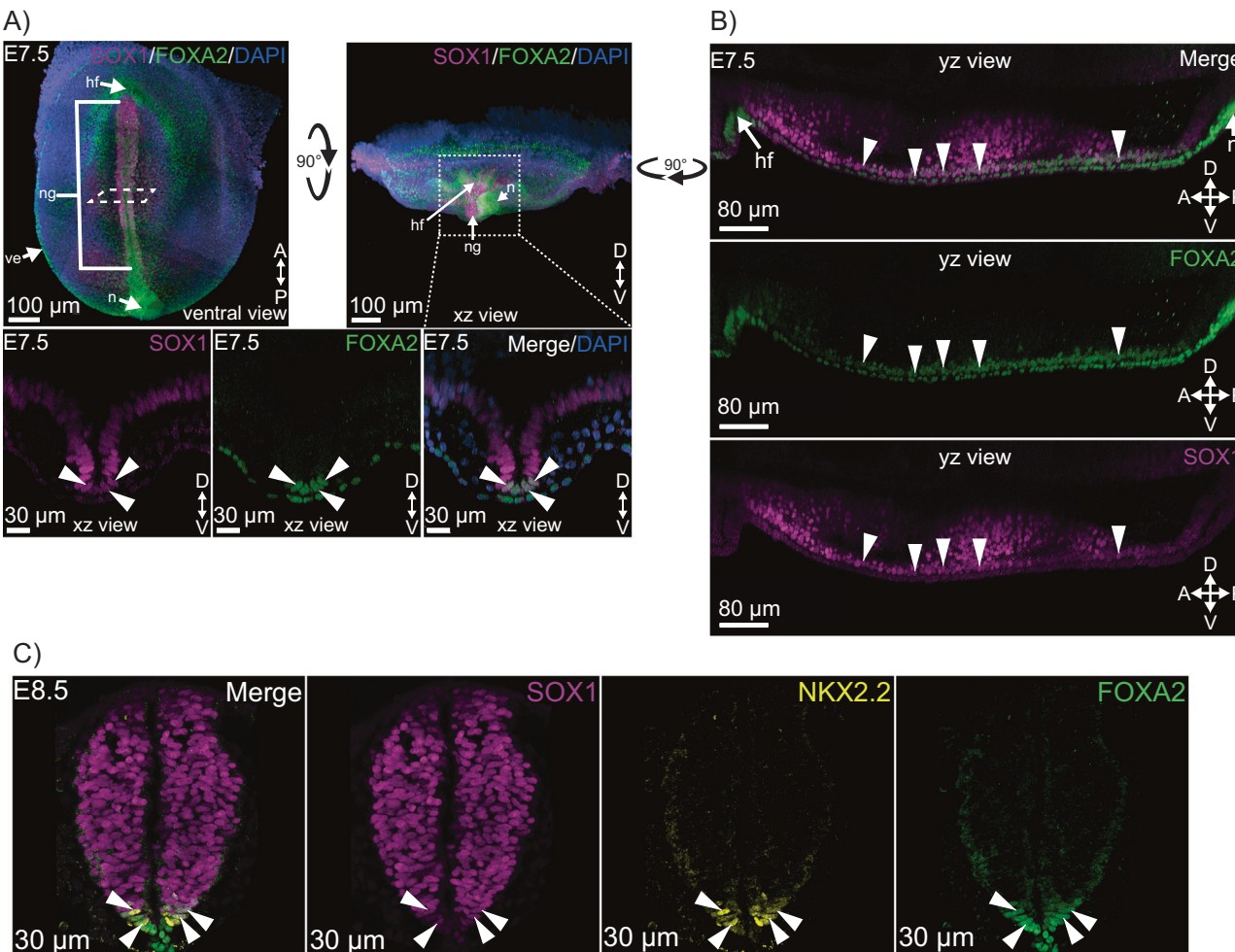

**Fig. 6 | Immunofluorescence staining of whole E7.5 embryos and E8.5 embryo neural tube slices confirms the presence of a SOX1+FOXA2+ population in vivo and suggests SOX1+FOXA2+ cells as precursors to p3 neural progenitors.**
**A** Representative image of E7.5 Embryo stained for SOX1 and FOXA2. SOX1 expression was detected in the neural groove and neurectoderm as expected[48]. Bottom images are optical cross sections of the neural groove when rotating the embryo 90 degrees around the left-right axis. FOXA2 expression was detected as expected in the visceral endoderm and node[46,47], and also in the ventral tip of the neural groove. SOX1+FOXA2+ cells were detected ventrally in the neural groove (marked with arrowheads). Representative images from 5 embryos shown. A:
anterior, D: distal, P: posterior, V: ventral. hf: head fold, n: node, ng: neural groove, ve: visceral endoderm. **B** Optical section along the anterior-posterior axis, cutting through the midline of the E7.5 embryo in A) shows that SOX1+FOXA2+ cells could be found along the whole length of the neural groove. **C** Representative image of neural tube slice from an E8.5 embryo stained for SOX1, FOXA2, and the marker for p3 neural progenitors NKX2.2. SOX1 expression was detected in the neural tube as expected[48] and FOXA2 expression was detected in the floor plate and notochord as expected[46,47]. SOX1+FOXA2+ cells were still present at E8.5, located ventrally, and found to co-express NKX2.2[49], a marker for p3 neural progenitors. $N = 3$ embryos.

in young and old embryos ($47.4 \pm 20.3\%$ and $44.3 \pm 12.5\%$ respectively). This suggests a developmental trajectory where SOX1+FOXA2+ cells at E7.5 are the progenitor population for p3 neural progenitors and upon p3 fate acquisition, these cells lose SOX1+FOXA2+ expression.

## Discussion

Using timelapse microscopy followed by IMC, we gained multi-dimensional snapshot protein expression data combined with cell kinship, lineage and NANOG dynamics history of mouse embryonic stem cells in pluripotency and differentiation media. With this infor-mation, we attempted to decipher the transition states of single cells from pluripotency to germ layer marker expression. Cell states based on protein expression were assigned to cells at the end of the obser-vation, and their history was analyzed over several generations. Con-sistent with previous studies, we found the core pluripotency network of NANOG, SOX2 and OCT4 to be highly stabilized in pluripotency media[50]. While NANOG is downregulated in a small proportion of cells in pluripotency media, this is an unstable state, and cells revert back to

a NANOG+ state within 1-2 generations. We also observed that a variety of transcription factors and epigenetic regulators are expressed in diverse combinations in this pluripotent state. This suggests a stable network with redundancies where downregulation of one plur-ipotency factor is not an indicator of differentiation. This network is destabilized in RA differentiation medium, where the frequency of detected cell states over time suggests a downregulation first of Klf4 and Tfcp2l1 followed by downregulation of the core pluripotency network (Supplementary Fig. 2B).

Focusing on SOX1 as a canonical marker of the neuroectoderm lineage, we identified sister pairs where SOX1 was asymmetrically upregulated. This demonstrates the rapid commitment to a Sox1 expressing state in the very short time since the sisters' division. Looking at NANOG dynamics between these SOX1+ cells and SOX1-cells as well as their ancestors, we found no indication that NANOG downregulation alone is sufficient for SOX1 upregulation. Narrow-ing our analysis to SOX1+/SOX1- sister pairs in the final generation for which IMC data exists, we again found none of our IMC

quantified protein candidates to be differentially expressed within the sister pairs.

Our approach did identify a cell population expressing FOXA2, a canonical endoderm marker[34,51], during RA mediated differentiation. To our knowledge it had not been reported to be expressed in ESC cultures in ectoderm-promoting RA medium, presumably because it had not been assayed for due to classical immunostaining's limit to 3-4 simultaneous protein analyses. While XEN-like cells with characteristics of primitive endoderm have been generated from RA medium, we did not find their markers (SOX17, GATA6) to be co-expressed in our FOXA2+ cells[35]. Further, these XEN cells were reported only after 4 days of differentiation[36]. We already observe FOXA2+ cells as early as 1.5 days post RA differentiation, suggesting an early choice for cells between ectoderm and endoderm. Characterizing this FOXA2+ population by RNASeq, we determined that FOXA2 is an early marker in RA indicating visceral/parietal endoderm lineage choice, prior to the upregulation of other endoderm markers.

We also observed cells co-expressing the canonical uni-lineage ecto- versus endoderm commitment markers SOX1 and FOXA2, which we confirmed both by immunostaining and with live reporters. Unlike mesendoderm, where Brachyury is expressed prior to the rise of endoderm markers[37], SOX1+FOXA2+ cells are not a precursor to cells which express only either ectoderm or endoderm markers because singe-cell Sox1 and FOXA2 expression dynamics quantification revealed that cells can be Sox1-FOXA2+ and vice versa without first acquiring the SOX1+FOXA2+ fate. FOXA2 was upregulated post SOX1 upregulation but not vice versa. RNASeq revealed these SOX1+FOXA2+ cells to be in a unique state expressing both endoderm and neuroectoderm markers, and their progeny also displaying mixed morphology characteristic of both neuroectoderm and endoderm, suggesting bi-potency. Quantification of Sox1 and FOXA2 dynamics in live single cells revealed that ~60% of cells that express Sox1 go on to express FOXA2 (Fig. 4E). IMC and immunostaining experiments show that ~10% of SOX1+ cells are also FOXA2+ (Fig. 3C, G). Thus, a substantial number of cells have the potential to upregulate FOXA2 even upon SOX1 expression and enter a unique cell state. Sox1 is widely used as a canonical marker to identify neuroectoderm. However, our results caution us from using it as the sole marker for identifying neuroectoderm.

Assaying for these markers in whole post implantation embryos, we also found this population to be expressed as early as E7.5, in the ventral tip of the neural groove, in direct contact with the visceral endoderm. While FOXA2 immunostainings in mouse embryos exist, literature is focused on its expression in the node, notochord and endoderm[43], not in the neural groove and co-expression with SOX1. At E7, no SOX1 expression was observed, but within ~12 hrs, the entire neural groove is marked SOX1+, in line with the in vitro dynamics observed here. The position of these SOX1+FOXA2+ cells in-vivo suggests cells of the neural groove in direct contact with visceral endoderm acquire FOXA2+ expression. This is consistent with in vitro experiments where FOXA2 upregulation occurs post Sox1 but not the reverse. It is also consistent with lineage analysis that found SOX1+FOXA2+ cells to always be closely related to uni-lineage marker SOX1+ or FOXA2+ cells in vitro. At E8.5, we also found a SOX1+FOXA2+ cell population at the ventral tip of the neural tube, which is formed with closing of the neural groove, co-expressing NKX2.2, a marker for p3 progenitors. We hypothesize that these SOX1+FOXA2+ represent a unique cell state that later gives rise to p3 progenitor cells in vivo. This is based on 1) the specific location of SOX1+FOXA2+ cells at E7.5 and E8.5 and of SOX1+FOXA2+NKX2.2+ cells at E8.5; 2) the quantification that revealed a majority of NKX2.2+ to also be SOX1+FOXA2+, especially in the younger embryos of ~E8.5; and 3) the observed dynamics in-vitro. It has previously been hypothesized that the notochord (of mesodermal origin and expressing FOXA2) induces neural floor plate formation as well as specification of neural tube progenitors via the Shh morphogen[49,52,53]. Cells co-expressing NKX2.2 and FOXA2 in the neural tube had been identified in chick embryos but thought to be involved in floor plate formation in conjunction with the notochord[52,53]. However, more recently in mouse embryos, lineage tracing experiments have shown that p3 neural progenitors at E11.5 have a history of FOXA2 expression and display a unique epigenetic signature, distinct from other neural progenitor populations[49]. Our study now suggests that this population has its origin in the SOX1+-FOXA2+ cells observed at E7.5 and not from the notochord. This is because we observe FOXA2+ expression in the neural groove prior at E7.5 prior to the formation of the notochord, with its precursor, the node clearly marked by FOXA2+ and in a spatially different part of the embryo. This suggests that the observed SOX1+FOXA2+ population in vivo is a product of the interaction between neuroectoderm cells in close contact with visceral endoderm cells, consistent with SOX1+-FOXA2+ cells characterized in vitro. Thus, our study here identified a developmental cell state marked by SOX1 and FOXA2 co-expression.

## Methods
### Cell lines
All experiments were performed with mouse embryonic stem cells. All mouse lines are from R1 line[54] except where stated as NG4 (NANOG GFP, CCE line)[38] and SOX1-EGFP (46C line)[39]. Wild type and engineered lines were routinely maintained in standard SerumLIF conditions and incubated at 37 °C in 5% CO2. All lines were tested for mycoplasma and found to be negative.

### Mice for embryo isolation
Animals used were 15- 20 weeks old RjOrl:SWISS mice from Janvier Labs. Mice were housed in ventilated cages with 4-6 mice per cage. They were housed with an inverse 12 h day-night cycle (Temperature: 21 ± 2 C, Humidity: 55 ± 10 %) and with access to standard diet and water as required. Mice were visually inspected daily for signs of distress and pain by animal facility caretakers. For the timed matings, one male and one female per cage were paired. For staging the embryos, the afternoon of the day that a mating plug was observed, was considered embryonic day 0.5. Pregnant mice were euthanized by CO2 inhalation. Additionally, death was confirmed by cervical dislocation.

### Generating the NANOGVenus/iRFPnucmem reporter ESC line
R1 NANOGVenus cells[24] were stably transfected with iRFP labelled nucmem[27] under constitutive CAG promoter using PiggyBac transposon system. After 1 week, cells were FACS sorted for positive Venus and iRFP expression to obtain the desired cell line.

### Cell Culture
All ES lines were routinely cultured in SerumLIF media on 0.1% gelatin (Sigma-Aldrich, G1890-100G) coated plates. SerumLIF media consists of 10% FCS (PAN, P30-2602), 10 ng/ml LIF (Cell guidance Systems, GFM200-1000), 2 mM GlutaMAX (Thermo Fisher, 35050-038), 1% non-essential amino acids (Thermo Fisher, 11140-035), 1 mM sodium pyruvate (Sigma-Aldrich, S8636) and 50 µM β-mercaptoethanol (Sigma-Aldrich, M6250) in DMEM (Thermo Fisher, 11960-085) basal media. Prior to timelapse or immunostaining experiments, cells were seeded in slides (Ibidi, 80606) or 12 well chambers (Ibidi, 81201) coated with E-cadherin (Primorigen Biosciences, S2112-500UG). For coating, E cadherin was added at 0.4 ug/100 ul PBS++ per well/slide and incubated for 1 h at 37 C. For timelapse experiments with NANOGVenus/iRFPnucmem, media was exchanged to either SerumLIF, RA or Meso medium. The composition of SerumLIF is the same as in routine cell culture except basal media is phenol red free DMEM (Sigma-Aldrich, D1145). For RA and Meso medium, the basal medium is N2B27, self-made or commercial (Ndiff, Takara, Y40002). Self-made N2B27 consists of equal volumes DMEM/F12 (LifeTech, 21041025) and Neurobasal medium (LifeTech, 12348017) supplemented with 100 U/ml Pen/Strep (15140122, ThermoFisher), 2 mM Glutamine (G7513, Sigma-Aldrich), N2

(17502-048, LifeTech) and B27 (12587-010, LifeTech). For RA medium, Retinoic Acid (Sigma-Aldrich, R2625) was added at 0.1 uM to basal medium. Meso medium comprises of 100 ng/ul Activin-A (RayBiotech, 2281022) and 3 uM CHIR99021 (R&D, 4423) in N2B27 medium. For experiments testing co-expression of SOX1 and FOXA2 in R1WT and NG4 cells, RA was added at 0.1 uM, 0.5 uM or 1 uM to N2B27 basal medium as depicted in figures. For timelapse experiments with Sox1-EGFP/FOXA2mCherry line, the RA basal media is phenol red free Serum medium, i.e. SerumLIF minus LIF, to reduce auto fluorescence when imaging EGFP. To this RA added at 1 uM.

## Timelapse Microscopy

All timelapse experiments were done using a wide field Nikon Ti-E microscope with a Nikon 10x Plan Fluor objective, Nikon Perfect Focus Systems, Lumencor Spectra X light engine and Hamamatsu Orca Flash 4.0 camera using software Youscope v2.1[55]. Imaging frequency was 20–25 minutes for a duration of 46–48 hours. Fluorescent proteins were imaged using teal (Venus), cyan (EGFP), green (mCherry) and red (iRFP) excitation lights and with the following optimized filter sets (Excitation filter; Dichroic mirror; Emission filter): EGFP (470/40; 495LP; 525/50), Venus (500/20; 515LP; 535/30), mCherry (550/32; 585LP; 605/15) and iRFP (655/40; 685BS; 716/40).

## Imaging Mass Cytometry

Post timelapse imaging, cells were fixed for 30 minutes with 4% PFA (Sigma-Aldrich, HT5011-1CS) and quenched with 100 mM glycin (AppliChem, A1377) for 10 minutes. Cells were permeablized for 5 minutes in 0.2% Triton-X (AppliChem, A1388) and then incubated with blocking buffer for 1 hour. The composition of the blocking buffer was 5% donkey serum (Jackson Immuno Research, 017-000-121) + 0.1% Triton-X in PBS. Before continuing with IMC, cells were imaged again as last timepoint of timelapse movie to be used as reference in aligning timelapse and IMC images. Cells were incubated overnight at 4 °C in a single cocktail of the 37 metal-conjugated antibody panel resuspended in blocking buffer in a single cocktail (Supplementary Table 1). Cells were washed in TBS three times prior to staining with 0.5 µM Cell-ID Intercalator-Ir (Fluidigm, 201192B) for detection of DNA. After 5 min, slides were rinsed with TBS and then briefly in water and air dried.

Chamber slide walls were then removed and Multiplexed images of the TMA cores were acquired using a Hyperion Imaging System (Standard BioTools). A square area of the chamber slide well was acquired at 200 Hz, and the raw data were preprocessed using commercial software (Standard BioTools).

## Antibody panel for Imaging Mass Cytometry

An antibody panel was composed to include relevant markers implicated in pluripotency and differentiation including germ layer markers as well as signaling molecules, histone modifiers and cell cycle markers to gain a comprehensive view of cell state during ES cell differentiation. Antibodies were validated for expected expression under ESC differentiation conditions, cell compartment specificity, and signal intensity prior to metal-chelating polymer conjugation and further confirmation by IMC. The complete list of antibodies can be found in Table S1.

## Immunostaining of cultured cells

Cells were immunostained as described previously[56]. Briefly, cells were fixed for 10 min with 4% PFA (Sigma-Aldrich, HT5011-1CS), then 10 min with ice cold methanol (Sigma-Aldrich, 32213). 0.2% Triton X (AppliChem, 1388) for 15 minutes was used for permeablization followed by 2 h in acid bleaching solution to bleach fluorescent proteins. Bleaching solution consists of 20 mM HCl (Merck, 1.00317.1000) and 3% H2O2 (Sigma-Aldrich, H-1009) in PBS. Cells were incubated in blocking buffer for 1 h. Blocking buffer is composed of 0.1% Triton X and 5% donkey serum (Jackson ImmunoResearch, 017-000-121) in PBS. Primary antibody staining was done overnight at 4C. Secondary antibody staining, if

required, was done at room temperature for 1 h. Primary antibodies used were labelled anti-FOXA2_AF647 (1:400) (Abcam, ab193879), anti-NANOG_AF488 (1:200) (Ebioscience, 53-5761-80) and unlabeled goat anti-SOX1 (1:200) (R&D, AF3369). Donkey anti-goat AF555 (1:1000) (ThermoFisher, A-21432) was used for secondary antibody staining in the case of SOX1. Nuclei were labelled with DAPI. Snapshot immunostaining imaging was performed in the same setup as timelapse microscopy.

## Generating knock in reporter line

Sox1-GFP/FOXA2mCherry double reporter line was generated using Crispr-Cas9. gRNA to nick FoxA2 at 3' end was designed in Benchling and cloned into plasmid co-expressing Cas9. This was co-transfected into Sox1-EGFP cells along with plasmid containing mCherry flanked by FoxA2 homology arms at 3' end. After 1 week, cells were single cell sorted in 96 well u-bottom plates and expanded. Colonies were genotyped for the presence of mCherry and successful candidates underwent RA differentiation assay for 2 days. To ensure correct expression and localization of FOXA2mCherry, cells were imaged for mCherry and EGFP expression followed by immunostaining for FOXA2 and SOX1. mCherry expression was correlated with FOXA2 antibody staining expression at the single cell level and the successful clone was chosen based on highest degree of correlation between mCherry and antibody expression.

## Sample preparation for RNASeq and Bulk RNA-Sequencing

Sox1-EGFP/FoxA2mCherry cells were seeded in SerumLIF overnight in 0.1% gelatin coated 6 well plates (ThermoScientific, 140675) before medium exchange to commercial N2B27 (Takara, Y40002) + RA (1 uM). Cells underwent differentiation for either 2 days, 4 days or 6 days in this medium as indicated with fresh medium exchange every 2 days, prior to FACS sorting for desired populations. FACS sorting was done in a BD FACSAria III sorter with BD FACS Diva v8 software. Gates for sorting desired populations were set with R1WT cells as the negative control. Gating strategy as shown in Supplementary Fig. 5D was plotted with FlowJo v10. For samples where the cell state of the sorted population was to be analyzed by RNASeq, 500 cells per sample were sorted into low-binding 96-well plates (Eppendorf) filled with 1xPBS, flash frozen with dry ice and promptly stored at −80 °C until use. For samples where the progeny of sorted population was to be analyzed by RNASeq, 50000 cells were sorted per desired population post 2 days of differentiation and replated in 0.1% gelatin coated 48 well plates (Thermo Scientific, 150687) containing RA medium (1 uM). The basal medium for RA was Serum, i.e SerumLIF used in routine culture minus the LIF. Post expansion for a further 2 days or 4 days as indicated, cells were harvested, resuspended in PBS, concentration manually counted with a Neubauer chamber and ~500 cells per sample were transferred to low binding 96-well plates (Eppendorf) and stored immediately at −80 °C until use following a flash freeze with dry ice. RNA extraction from samples involving isolation and fragmentation, and the synthesis of the double-stranded cDNA library were performed according to the SMART-Seq Stranded Kit protocol from Takara (Catalog # 634444). The cDNA library was sequenced on the Illumina NovaSeq6000 platform.

## Embryo isolation and fixation

For the isolation of the embryos, the uterus of the pregnant mice was dissected and placed on a petri dish with PBS. The uterus was cut between the bulges formed by the embryos covered by the decidua (a layer of tissue that supports the embryo development). These pieces were then placed on the lid of a petri dish filled with PBS + 10% FCS. The embryos were isolated using jeweler's forceps and kept in PBS on ice until all embryos were isolated. Once the embryos were isolated, the staging was confirmed by morphology, using the Theiler stage system. Embryos were selected at random for fixation with no specific bias towards any sex. For fixation, E6.5-7.5 embryos were fixed for 30 minutes with 2% PFA at RT and then washed twice for 5 minutes in

PBS. E8.5 embryos were fixed for 1 hour with 4% PFA at RT and then washed twice for 20 minutes in PBS.

## E8.5 embryo cryosection

Fixed E8.5 embryos were incubated O/N, at 4 C, in PBS 15% Sucrose. Embryos were then incubated again O/N, at 4 C, in PBS 30% Sucrose. Embryos were then transferred to a cryomold and covered with Poly-Freeze Tissue Freezing Medium (SIGNA, SHH0026-120ML). Cryomolds with embedded embryos were then placed at −80 C for freezing. Embryos were then sliced at a thickness of 16 μm on a cryostat at −20 C and placed on Superfrost® Plus slides (VWR, J1800ABDH). Slices were washed once prior to staining, for 15 min at RT, in PBS.

## E6.5-E7.5 and E8.5 neural tube slices immunostaining

Embryos/slices were incubated for 1 hour at RT in Block/Perm solution (PBS with 10% NDS and 0.5% Triton X). Afterward, embryos/slices were incubated in Block/Perm solution with primary antibodies, O/N, at 4 C. Embryos/slices were then washed 3 times for 15 minutes, at RT, in PBS with 0.05% Triton X. Embryos/slices were then incubated in Block/Perm (with 0.05% Triton X) with secondary antibodies for 1.5 hours, at RT. Embryos/slices were then washed 3 times for 15 minutes, at RT, in PBS with 0.05% Triton X and one last time for 15 minutes in PBS. The primary antibodies used were the following: rabbit anti-FOXA2 (abcam, ab40874, 1:300 dilution), goat anti-hSOX1 (R&D, AF3369, 1:100 dilution), mouse anti-NKX2.2 (DSHB, 74.5A5, 1:100 dilution). The secondary antibodies used were the following: donkey anti-mouse IgG (H + L) Alexa Fluor™ 488 (Thermo Fisher, A-21202, 1:500), donkey anti-mouse IgG (H + L) Alexa Fluor™ plus 647 (Thermo Fisher, A32787, 1:500), donkey anti-rabbit IgG (H + L) Alexa Fluor™ 488 (Thermo Fisher, A-21206, 1:500), donkey anti-rabbit IgG (H + L) Alexa Fluor™ 546 (Thermo Fisher, A10040, 1:500), donkey anti-goat IgG (H + L) Alexa Fluor™ 546 (Thermo Fisher, A-11056,1:500), donkey anti-goat IgG (H + L) CF640R (Biotium, 20179,1:500).

## Two-photon excitation microscopy

Two-photon excitation (2PE) microscopy was performed to image whole E6.5-E7.5 embryos. Samples were placed on glass bottom 35 mm dishes (Matek, P35G-1.5-14-C) and embedded in 0.5% low melting point agarose (Promega, V2111). E7.5 embryos were placed on the dish so that the neural groove faced the glass. Microscopy was performed on a Zeiss LSM980 microscope equipped with a tunable INSIGHT X3 680-1300 NLO laser, 2 PMT, and 2GaAsP non-discanned detectors, using a 40X C-Apochromat water objective (NA 1.1, WD 0.62 mm), and Immersol® W 2010 (Zeiss). The scanning was performed at the highest LSM scan speed, with 2x averaging, in bidirectional mode at 1024×1024 pixel resolution, 0.6−0.8X zoom, and a z-step size of 0.410 μm. Auto Z Brightness correction was used and the following laser and detector settings were used: 760 nm at 0.8-16% laser power (LP), PMT detector at 600 V (for DAPI); for 930 nm at 2−18% LP, PMT detector at 650 V (for Alexa Fluor 488); 840 nm at 0.8−16% LP, GaAsP detector at 600−650 V (for Alexa Fluor 546); 830 nm at 1−16% LP, GaAsP detector at 650 V (for CF640R).

## Confocal microscopy

Confocal microscopy was performed to image the E8.5 neural tube slices. Samples were incubated for 5 min in 25%, 50% and 75% Glycerol diluted with PBS, and finally covered with glycerol mounting medium (80% Glycerol (BioChemica, A1123,1000), 0.1 M NPG (SIGMA, 02370-100G) in TBS) and a #1.5 glass coverslip (TED PELLA, 260146). Microscopy was performed on a Leica SP8 confocal equipped with 405 nm, 561 nm, 633 nm diode lasers, 488 nm Argon laser, three photo-multiplier tubes (1 used for DAPI), two HyD detectors (used for Alexa Fluor 488, plus 488, 546, plus 647, and CF640R) using a 63x HCX PL APO CORR CS glycerol immersion objective (NA 1.3, WD 0.28 mm) with Leica type G immersion liquid. Microscopy was also performed on a Leica SP8 Falcon confocal, equipped with the same lasers and

detectors as the Leica SP8, but with an additional white light laser (WLL), using a 63x HC PL APO CS2 glycerol immersion objective (NA 1.3, WD 0.3). with Leica type G immersion liquid. The scanning was performed at 400 Hz, in bidirectional mode, with 2x averaging at 1024 × 1024 pixel resolution, 16-bit, 0.75-0.9 zoom, and a z-step size of 0.5 μm. Laser and detector settings were: 405 nm laser at 0.5% Laser power (LP), PMT at 600−700 V gain (for DAPI), 488 nm laser at 1−4% LP, HyD at 20%−40% gain (for Alexa Fluor 488 and, plus 488), 561 nm laser at 2−6% or WLL at 556 nm and 2-5% LP, HyD at 20−30% gain (for Alexa Fluor 546), 633 nm laser at 3−10% or WLL at 642/647 and 3−8%, HyD at 20−40% (for CF640R and Alexa Fluor plus 647).

## Processing and analysis of IMC data

Acquired IMC data was segmented on Ilastik v0.5 using a random forest classifier of cell nuclei, cytoplasm and background and quantified on histoCAT[57] v1.0. Downstream analysis was performed in R using pre-existing functions and custom scripts adapted from Nowicka et al.[58] Umap was generated using "umap" package following range scaling of the expression data from all antibodies in the panel. To elaborate, per antibody, from individual values the 1 percentile value was subtracted. This was then normalized to the value obtained by subtracting the 1 percentile value from the 99-percentile value for that antibody distribution. This is called 'min-max' normalization such that all values of a distribution are scaled to a range of 0-1. Any values greater than 99 percentile or less than 1 percentile were set to 1 and 0 respectively. The parameters included all expression data from the antibodies panel. Distributions to fit antibody density curves were produced using "mixtools" and "fitdistrplus" packages.

## Processing timelapse data

16-bit images acquired during timelapse imaging were converted to 8 bit and corrected for background inconsistencies using BaSiC[59] v1.1. Cell nuclei were segmented using FastER v1.4 with iRFPnucmem as segmentation channel[60]. Cells were manually tracked using TTT v3.5 and fluorescence expression quantified using QTfy[25] v1.1. Cell lifetime, motility, cell generation were calculated from tracked tree data using custom code in C++ and linked to timelapse data. Further downstream analysis was done in R. SOX1EGFP and FOXA2mCherry fluorescence traces were plotted in GraphPad Prism v10.1.2.

## Combining timelapse + IMC data sets

IMC images acquired at the end of timelapse movies were rotated and aligned to time lapse frames with 'Image Registration' in Matlab R2017B. Post rotation x and y coordinates of cells in IMC dataset were noted. Then Manhattan distance between last cells of tracked time-lapse data and cells of IMC data was calculated per field of view to account for minute shifts in frames between timelapse and IMC data. Cells with the least Manhattan distance between the two datasets, with a maximum allowed shift of 30 pixel, were linked together. A subset of linked cells were checked manually to ensure that the process had worked correctly without spurious links.

## Analysis of immunostaining data from cultured cells

16-bit immunostaining images were converted to 8 bit and background corrected in BaSiC[59] v1.1. If required, images were aligned to previous frames, e.g. from timelapse data, using 'Image Registration' in Matlab R2017B prior to segmentation. Cells were nuclear segmented in DAPI channel and fluorescence signal quantified in FastER[60] v1.4. Downstream data analysis was performed in R (version 4.1.2).

## Embryo image visualization and analysis

Tile scans of confocal microscopy were stitched directly after acquisition using the Leica LAS X software. Tile scans of 2 photon excitation (2PE) microscopy were stitched using the stitcher of Huygens Software (Scientific Volume Imaging). Image visualization and, where applicable,

cropping from tile scans were done in Imaris v10 (Bitplane). Confocal neural tube slice images and 2PE pe whole embryo images are Imaris 3D reconstructions using the MIP (maximum intensity projection) mode. Optical sections of E7.5 embryos were done with the oblique slicer in Imaris, at minimal thickness.

## RNASeq analysis

The obtained reads were aligned to the M 21 release mouse genome (ftp://ftp.ebi.ac.uk/pub/databases/gencode/Gencode_mouse/release_M21) using STAR[61]. Subsequently, sorted.bam files were generated using SAMtools[62]. Counts were then extracted using featureCounts[63]. Subsequent analysis of the resulting count matrix was performed in R (v4.1.2). Normalized counts were obtained and PCA analysis done with R package "DESeq2"[64] by following standard analysis pipeline (https://bioconductor.org/packages/devel/bioc/vignettes/DESeq2/inst/doc/DESeq2.html). Heatmap generation was performed on normalized counts using R package "pheatmap".

## Inferring NANOG Transition Dynamics

Kin Correlation Analysis (KCA) was performed as described previously[29] to infer NANOG transition dynamics per generation using end state and lineage data with assumption that cell state is reversible. Briefly, based on NANOG IMC threshold, last cells of timelapse trees were classified as N+ or N-. Mean transition probability was calculated as the mean of transition rates derived from lineage distance $u = 1$ (sisters) to $u = 5$ (distant cousins). A total of 51 trees were used in the analysis with bootstrapping ($n = 100$). Correlation matrices were constructed per u to count the instances of cells pairs which were N+/N+, N-N- or N+N- and normalized to the total count. Matrices were then normalized per column by the column sum, which is the steady state distribution of N+/N- cell states absent of lineage information. Finally transition probabilities per generation were calculated with the formula $T(u) = C(u)^{(1/2u)}$.

## Actual NANOG transition dynamics

Per cell, the first two timepoints post cell division and the last two timepoints pre cell division were excluded to eliminate technical noise. Based on NANOGVenus reporter expression, manual threshold was set classifying cells as N+ or N- per time point. For each tree, cells were assigned overall as N+ or N- if they were consistently N+ or N- for at least 5 consecutive time points (~2.5 hours) pre cell division. For each transition state, the number of occurrences were calculated based on current cell state and cell state post cell division and normalized to the total number of transitions to get transition probability. Mean transition was obtained by bootstrapping ($n = 100$) on 51 trees in SerumLIF, 70 trees in RA and 96 trees in Meso used for analysis.

## Determining decision cells

Final nodes/cells of tracked trees were assigned as Fate+ or Fate- (e.g. SOX1+ and SOX1-) based on manual threshold of IMC data. If sister cells had the same Fate+ state, trees were then recursively traversed up with upstream nodes assigned the same cell state as downstream nodes till a difference was observed in sister cell fate. The sister cell that is Fate+ at that level was marked as a 'Decision Cell'. Apoptotic cells were treated as having the same state as their sister cell, as the end composition is a homogenous colony. Ancestors of Decision Cells were assigned N+ or N- as described above to determine role of ancestor NANOG expression in lineage acquisition.

## Statistical analysis

Each experiment was independently replicated at least 2 times as indicated in figure legends. Statistical significance was determined by two-tailed unpaired student t test, two-way ANOVA or Fischer's exact test as indicated in figure legends and calculated using GraphPad Prism v10.1.2. P values below 0.05 were considered statistically significant. No statistical methods were used to pre-determine sample size. The experiments were not randomized, and investigators were not blinded to allocation during experiments and outcome assessment.

## Reporting summary

Further information on research design is available in the Nature Portfolio Reporting Summary linked to this article.

## Data availability

Processed imaging data from timelapse and Imaging Mass Cytometry has been uploaded to ETH Research Collection with https://doi.org/10.3929/ethz-b-000688864. The raw imaging data (too big for this repository) can be obtained from the corresponding author Timm Schroeder (timm.schroeder@bsse.ethz.ch). Raw images will be shared electronically within two weeks of request. Bulk RNASeq data has been uploaded to the Gene Expression Omnibus (GEO) database under accession code GSE259317. The cell lines and plasmids generated in this study are available on request from the corresponding author Timm Schroeder. Source data are provided with this paper.

## Code availability

Custom R code used to analyze combined timelapse and Imaging Mass Cytometry data has been uploaded to ETH Research Collection with https://doi.org/10.3929/ethz-b-000688864.

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

## Acknowledgements

We thank Konstantinos Anastassiadis for kind gift of SOX1EGFP line, Heiko Lickert for FOXA2mCherry knock-in plasmid, and Sadanand Hormoz and Michael Elowitz for Kin Correlation Analysis code.

## Author contributions

Conceptualization, S.S., H.W.J., B.B., T.S.; Methodology, S.S., G.A., H.W.J., D.S., S.E., S.H., D.C., M.A., G.C.; Investigation, S.S., G.A., H.W.J., D.C.; Data curation, G.A., M.A.; Formal Analysis, G.A., D.C., M.A.; Software, S.S., G.A., D.S.; Visualization, G.A., T.S.; Supervision, G.C., B.B., T.S.; Resources, A.R., D.L.; Writing – Original Draft, G.A., T.S.; Project administration, B.B., T.S.; Funding acquisition, B.B., T.S.

## Funding

## Competing interests

The authors declare no competing interests.

### Ethics

All experiments were performed according to Swiss federal law and the institutional guidelines of ETH Zurich as well as approved by local animal ethics committee of Basel-Stadt (approval number 2655).

## Additional information

¹Department of Biosystems Science and Engineering, ETH Zurich, Basel, Switzerland. ²Department of Quantitative Biomedicine, University of Zurich, Zurich, Switzerland. ³Institute of Molecular Health Sciences, ETH Zürich, Zürich, Switzerland. ⁴Present address: Department of Neurosciences, Leuven Brain Institute (LBI), KU Leuven – University of Leuven, Leuven, Belgium. ⁵Present address: Laboratory of Neurobiology, VIB Center for Brain & Disease Research, Leuven, Belgium. ⁶Present address: Lunenfeld Tanenbaum Research Institute, Mount Sinai Health Systems; Department of Molecular Genetics, University of Toronto, Toronto, ON, Canada. ⁷Present address: Institute for Computational Biomedicine, Heidelberg University, Faculty of Medicine, Heidelberg University Hospital, Heidelberg, Germany. ⁸Present address: Institute of Pathology, Heidelberg University Hospital, Heidelberg, Germany. ⁹Present address: Translational Spatial Profiling Center (TSPC), Heidelberg, Germany. ¹⁰Present address: Department of Hematology, St. Jude Children's Research Hospital, Memphis, TN, USA. ¹¹Present address: Department of Pathology and Laboratory Medicine, The University of Tennessee, Memphis, TN, USA. ✉e-mail: timm.schroeder@bsse.ethz.ch

