## [Peer Review File · Nature Communications]

REVIEWER COMMENTS

Reviewer #1 (Remarks to the Author):

In this study, the authors use time-lapse imaging to study the expression of Nanog over 48 hours in mouse embryonic stem cells (ESCs) cultured under 3 different medium conditions. They combine this with endpoint Imaging Mass Cytometry for a panel of 37 pluripotency, lineage-associated, and epigenetic markers.

The authors main conclusions from this study are:

- 1) Nanog reporter downregulation does not correlate with neurectoderm lineage specification.
- 2) A rare population is observed in the presence of RA medium that coexpresses SOX1 and FOXA2. The authors suggest that this population represents a bipotential neurectoderm/definitive endoderm progenitor.

Major concerns:

1. In these experiments, only 2 biological replicates were analyzed. Typically, 3 biological replicates are the minimum for any inferential analysis. Moreover, it is unclear why one set of samples was 'Discarded' (Fig. S1A). The figure legend states that this was an experiment using different antibodies. Does this mean an entirely different set of markers or a different batch of the same antibodies? This needs to be further explained.

2. The authors state that these experiments were performed to better understand the transition steps as cells lose pluripotency and differentiate into various germ layers and that this requires continuous information of an individual cell's state at any time point. However, the experimental design here only studies Nanog reporter expression dynamics, and all other measurements are carried out at an endpoint. Moreover, several studies have previously analyzed Nanog dynamics under different medium conditions [1-4], and the study here did not generate any new insights based on these data on early ESC lineage specification. Thus, the purpose/utility of Nanog tracking here is unclear.

3. In RA medium, around 1% of cells coexpressed SOX1 and FOXA2. The authors hypothesize that these cells represent bipotential neurectoderm/definitive endoderm precursors. The current manuscript states that "We identify a novel developmental cell population and find unexpected lineage plasticity..." However, there is no attempt to further examine these cells and thus this statement is not supported. Since this is one of the main conclusions of this manuscript, further characterization of these cells is necessary:

- Specifically, while the authors use FOXA2 as a definitive endoderm marker, it is also expressed in other lineages e.g. extraembryonic primitive endoderm, visceral endoderm (including anterior visceral endoderm), cardiac progenitors [5], floor plate, axial mesoderm, and dopaminergic neurons. The authors should sort the SOX1- FOXA2+, SOX1+ FOXA2-, and SOX1+ FOXA2+ cells and perform in-depth transcriptional characterization to distinguish between these possibilities and particularly to understand what the FOXA2-expressing cells represent. Ideally, this would be sequencing but, at the least, a comprehensive panel of markers that can clearly distinguish between each of these lineages.
- The authors should sort individual SOX1+FOXA2+ cells and show that they can give rise to both

endoderm and neurectoderm (based on comprehensive marker analysis).

- In Fig. 3F, the authors show scatter plots of NANOG vs. FOXA2 and NANOG vs. SOX1 expression levels. Since the focus of these figures is the FOXA2+ SOX1+ cells, the authors should also show the same type of scatter graph of FOXA2 vs. SOX1 expression so that the double positive population can be clearly visualized in a quantitative manner.

- Importantly, no attempt is made to determine whether this SOX1+ FOXA2+ population exists in the embryo in vivo. As such the authors cannot claim to identify a “novel developmental cell population”. The authors should examine this either by antibody staining and/or chimera generation using their double reporter cell lines.

4. The authors generate a SOX1 FOXA2 double reporter cell line. However, the half-life of the specific fluorescent reporters used will confound the identification of a true SOX1+ FOXA2+ population. The authors should discuss this and at least compare the proportion of their cultures that are double positive using their reporter compared to their antibody staining. Moreover, the authors should perform a flow cytometry time course of their double reporter cell line in RA culture.

5. In Fig. 2, the authors compare the Nanog expression state of ‘Decision cells’. The conclusion is that 1) most decision cells are Nanog-, and 2) the % of Nanog+ cells is higher in earlier Decision cell ancestral generations. This is not surprising since most cells in RA cultures as a whole are Nanog- and the proportion of Nanog+ cells decreases over time, therefore earlier generations will have fewer Nanog+ cells across the entire culture. Although the authors compare the proportions of Nanog+ and Nanog- cells in Sox1+ vs Sox1- cells and their ancestors, they do not say whether the differences are statistically significant. If they are not statistically significant, this data is not informative regarding neurectoderm specification. Another caveat to this analysis is that based on the nomenclature of “Decision cells”, it seems the authors consider Sox1- cells as undifferentiated or not having made a “lineage decision”. At least a fraction of the Sox1- cells may also have differentiated but toward a non-neurectoderm lineage and hence Nanog dynamics may be similar between Sox1- and Sox1+ differentiating cells.

Additional questions and concerns:

1. Fig 1: the fluorescence level of the Nanog reporter increases around the time of cell division. Is this an artefact of the change in cell morphology at this time and, if so, is there some normalization that is needed at the time of cell division? The authors should discuss this.

2. The authors indicate that the probability of a cell transitioning from a Nanog- to a Nanog+ state is much higher than the other way around. This is surprising since Nanog is a marker of the pluripotent state and therefore it might be expected that it is more common for Nanog expression to be lost (i.e. through spontaneous differentiation) than gained. The authors should discuss this observation.

3. In Fig. 2D, the meaning of “cell lifetime” is unclear. Does this refer to actual cell lifetime i.e. time before cell death or time before cell division, or a combination of both of these? This should be stated more explicitly in the text.

4. In Fig. 4C, the authors should also provide overlay images.

5. The Discussion section states that since there was no difference in the 37 markers analyzed between SOX1+ and SOX1- sisters that there may be no transition steps between loss of pluripotency and neurectoderm differentiation. Previous studies of early ESC differentiation have already demonstrated a

dismantling of the pluripotency network within the first 25 hrs of differentiation in a medium permissive for neural specification [6] and thus this speculation is incorrect and should be removed.

1. Abranches, E., et al., Stochastic NANOG fluctuations allow mouse embryonic stem cells to explore pluripotency. *Development*, 2014. 141(14): p. 2770-9.
2. Pezzarossa, A., et al., Imaging Pluripotency: Time-Lapse Analysis of Mouse Embryonic Stem Cells. *Methods Mol Biol*, 2016. 1341: p. 87-100.
3. Hastreiter, S., et al., Inductive and Selective Effects of GSK3 and MEK Inhibition on Nanog Heterogeneity in Embryonic Stem Cells. *Stem Cell Reports*, 2018. 11(1): p. 58-69.
4. Jin, K.X., et al., N6-methyladenosine (m(6)A) depletion regulates pluripotency exit by activating signaling pathways in embryonic stem cells. *Proceedings of the National Academy of Sciences of the United States of America*, 2021. 118(51).
5. Bardot, E., et al., Foxa2 identifies a cardiac progenitor population with ventricular differentiation potential. *Nature Communications*, 2017. 8.
6. Kalkan, T., et al., Tracking the embryonic stem cell transition from ground state pluripotency. *Development*, 2017. 144(7): p. 1221-1234.

Reviewer #2 (Remarks to the Author):

The manuscript presented by Arekatla et al. combines cell tracking with high-dimensional protein profiling by Imaging Mass Cytometry (IMC) to simultaneously study the dynamics of ESCs differentiation and characterize their molecular features at the protein level. The approach employed is interesting and is within the continuing trend of connecting lineages, cell fate choice and single-cell transcriptomics/proteomics. The authors first perform time lapse microscopy to lineages and NANOG dynamics across 5-6 generations during 46 h differentiation of ESCs and IMC at the end to collect 37 proteins data. Then they evaluate NANOG expression and dynamic patterns and correlate its behavior with SOX1 (neuroectoderm lineage marker) expression patterns. They observe that NANOG is downregulated within 2-3 generations but is not required to for neuroectoderm cell fate choice. Then they found cells co-expressing SOX1 and FOXA2 (definitive endoderm marker), generate double reporter cell line to evaluate their expression dynamics during neuroectoderm differentiation. While the approach is interesting, the data and conclusions suffer from short differentiation time and data is underutilized. The observation of SOX1 and FOXA2 cells is interesting but validation of their molecular nature and functional potential is not resolved. It would have been useful if authors substantiated solid evidence to the set of observations presented, towards gaining new significant insights.

1) The differentiation time used (46 h) in the study is rather quite short to establish exclusive germ layer cell fates. While SOX1 expression can be observed by 46 h, its expression level as well as the proportion of cells are meaningfully distinct at later time points (PMID: 12524553). As a result, often 72 – 96 h is often used to differentiate ESCs to germ layer precursors. The following in the data make this reviewer suspect whether the differentiation time used is optimal.

- A) In Fig 1C, SOX1 levels is relatively high in cells of SerumLIF (SL) than in RA, the condition used to induce neuroectoderm differentiation.
- B) Similarly in Fig S1B, Brachyury levels in SerumLIF (SL) is relatively same as in Meso, the condition used to induce mesendoderm differentiation.
- C) In Fig 1D with dimensionality reduction, distinct differentiating subpopulations did not emerge. That the majority of pluripotency transcription factors and other regulators measured by IMC are still highly expressed suggests that there may not have been sufficient time for cells to segregate into distinct germ layer lineages.
- D) The potentially interesting role of FOXA2⁺ and FOXA2⁺ SOX1⁺ cells remain unresolved.

2) Majority of the cells in IMC image (Fig 1B) appear to be expressing SOX1. It is surprising given the 46 h differentiation time and that authors themselves found about 9% of SOX1⁺ cells (Fig 3C). A clear comparison of lineage markers signal observed in IMC, antibody staining and live-cell reporter is required to demonstrate the specificity of the antibody-isotope conjugates. It is understood that SOX1 levels are thresholded as in Fig 1E for subsequent analysis. However, it is key to show that the IMC reagents for lineage markers truly reflect their cell fate specific expression.

3) Even though this represents an interesting data set, a total of 37 protein panel consists of multiple lineage markers (SOX1, BRACHYURY, GATA6, OTX2) and two different differentiation conditions, the data remains underutilized with most analysis restricted to SOX1 expression patterns. How the other markers and their associated lineage patterns vary is not evaluated. Within the observed SOX1 patterns, it is also not clear how the patterns of all proteins vary and whether they can explain the cell state phenotypes.

4) Unbiased choice of thresholds based on gamma distribution fit is interesting and appears to be robust. Did the authors test similar unbiased thresholds for lineage markers such as SOX1, instead of the current choice from what appears to be more subjective? Also, if we go by the expression profile of SOX1 in Fig 1C, in contrast to results in Fig 1E is it likely that there would be more SOX1⁺ cells in SL than in RA?

5) “Nanog downregulation...is not sufficient for SOX1⁺ lineage commitment” (line 147). It is not surprising and is an expected result. NANOG is known to be downregulated at faster rate than other pluripotency factors during ESCs differentiation, resulting in dismantling of pluripotency network. While NANOG low cells are in general more prone to differentiation, it is not expected to suppress or promote germ layer precursor cell fates. However, NANOG (along with OCT4) is known to promote further downstream cell fates such as definitive endoderm (PMID: 21245162). Longer time differentiation analysis combined with perturbation tests would have revealed new insights for NANOG’s role in differentiation.

6) Observation of the unexpected SOX1⁺ FOXA2⁺ cells is interesting. However, the evidence provided in the paper does not resolve whether they represent “novel” differentiation stage as precursors for neuroectoderm and definitive endoderm lineages. The evidence provided is mostly observational and solid evidence is required to validate the hypothesis. For instance: i) verifying FOXA2 expressing cells with other definitive endoderm markers; ii) perturbation tests to validate the novel differentiation stage model; iii) long-term differentiation analysis in combination with lineage tracing, IMC and perturbation tests; iv) clear demonstration of their functional differentiation ability (at least in-vitro).

Other minor points:

7) Is the differentiation time 48 h or 46 h? In Fig 1A it is 48 h but the in rest of the text it is 46 h.

8) Is the panel of 37 antibody-isotope reagents custom generated by manual conjugation? If so how do these reagents act as novel resource compared to the existing CyTOF reagents to study ESC regulatory factors?

9) Which isotope is conjugated to which antibody? Specific list of isotopes used should be listed as part of Supplementary table 1.

10) In Fig S1B, what does 'pHOSPHO' and 'TFE3' represent? TFE3 is not in Table 1 and pSMAD1/5 distribution profile is not in Fig S1B.

Reviewer #3 (Remarks to the Author):

Overview: Authors developed an approach for live cell fate tracking with an end-point Imaging Mass Cytometry measurement. This approach allows to track to create cell pedigrees and study end-point cell fate distribution among kin cells, as well as to correlate it with cell's life cycle and motility. This approach also allows to calculate cell fate transition probabilities informed by fluorescence of a live reporter (in this case fused to Nanog protein). Authors also engineered a reporter cell line expressing fluorescent SOX1 and FOXA2, which allowed them to show mutual dynamics of these transcriptional factors.

The three main findings of the paper in my opinion are:

- SOX1 expression is preceded by Nanog downregulation in the cell's pedigree; however, Nanog downregulation alone is not sufficient for SOX1 cell fate,
- SOX1+FOXA+ cells are not an intermediate for SOX1+ or FOXA+ cell fates, these states exist independently,
- SOX1 expression always precedes SOX1+FOXA+ cell fate and never vice versa.

Using the platform established in this paper opens endless possibilities to study other known transcription factor dynamics in various directed differentiation protocols.

Minor suggestions/questions:

1. Fig 1B - clarify in the legend that you marked the example cell and its progeny with red circles.
2. Fig 1 C - it took me a moment to realize it's a histogram (a distribution?) of values for each protein. May be clarify that in the figure description.
3. Did you try to cluster the cells by the "combinatorial thresholded protein expression state"? There can be 2^{37} total cell fates based on binary thresholding for each of the 37 proteins in IMC, but may be some cell fates are preferred? This could reveal a variety of preferred cell fates.
4. Figure 1D: does excluding pHistone and a couple of other "outlier" proteins from analysis improve things?
5. Line 117: Are the transition probabilities different for different media conditions?
6. Line 118: why is it obvious that NANOG+ to NANOG- probability is lower than vice versa? NANOG is a pluripotency marker - why would a NANOG- transition to become NANOG+, thus "gaining" pluripotency?

Usually, cells tend to lose pluripotency during prolonged culture.

7. Fig 2 A, B - why does generation 0 in RA condition already have skewed population towards low Nanog? When was the timepoint for generation 0?

8. Line 140: could you calculate some metric to prove this statement more stringently? For example, compare raw NANOG levels on the -2 generation level for SOX1+ and SOX1- decision cells and do a t-test and show p-value here? This would greatly strengthen this statement.

9. Fig 2G - is it possible to zoom in on the violin plots along the y-axis? You can barely see any difference in their means at this scale.

10. Line 153: how do you define cell lifetime for the last generation cells, where the endpoint IMC measurement was made? What if cells would proceed to live longer if the endpoint experiment was not done? This makes the lifetime measurements inaccurate in that case.

11. Line 207: It is hard to see the light purple color (SOX1+FOXA2+ cells) in the figure 3G, since it is only 1% of the population. Therefore, not sure if referencing the Fig 3G for proof of SOX1+FOXA2+ presence makes sense. Either change the representation on Fig 3G or don't reference it here.

12. Where do the negative values in Fig 4B come from? If the plots are in log scale it needs to be indicated in the legend.

13. Line 273: how do you explain FOXA2 expression in your experiment then, since no one before observed it in similar experiments?

REVIEWER COMMENTS

Reviewer #1 (Remarks to the Author):

In this study, the authors use time-lapse imaging to study the expression of Nanog over 48 hours in mouse embryonic stem cells (ESCs) cultured under 3 different medium conditions. They combine this with endpoint Imaging Mass Cytometry for a panel of 37 pluripotency, lineage-associated, and epigenetic markers.

The authors main conclusions from this study are:

- 1) Nanog reporter downregulation does not correlate with neurectoderm lineage specification.
- 2) A rare population is observed in the presence of RA medium that coexpresses SOX1 and FOXA2. The authors suggest that this population represents a bipotential neurectoderm/definitive endoderm progenitor.

Thank you for your time and feedback!

Major concerns:

1. In these experiments, only 2 biological replicates were analyzed. Typically, 3 biological replicates are the minimum for any inferential analysis.

Thank you. Typically, three biological replicates are used where inferences are made at a population level, with the assumption that said population is homogeneous, so that any statistical analysis has a minimum of 3 replicate values to robustly estimate mean, standard deviation etc. In this study, with the IMC experiments, the goal was to look at heterogeneity within the analyzed population and deduce why some cells commit to certain lineages while their sister cells/ cells in e.g. the same neighborhood do not. Each cell thus represents an independent measurement and is a unique data point. All analysis in this manuscript utilizing measurements made at single cell level have sufficient numbers to make inferences and perform statistical tests. We can consider our IMC experiments to be analogous to single cell RNASeq experiments where we have often found studies to have 2 biological replicates even in recent studies from 2023, as the actual comparisons are between individual cells (Heezen et al., 2023; Múnera et al., 2023; Takada et al., 2023). Where inferences are to be made at population level, like with the new experiments in this manuscript using bulk RNASeq, we have included 3 biological replicates.

Moreover, it is unclear why one set of samples was 'Discarded' (Fig. S1A). The figure legend states that this was an experiment using different antibodies. Does this mean an entirely different set of markers or a different batch of the same antibodies? This needs to be further explained.

The antibodies used in the removed experiment are the same as the other two replicates. However, the cells in the discarded samples detached partially post fixation and prior to image acquisition leading to artefacts in the images. Thus, their position in the MDS plot represents the variation observed when the issue is of a technical nature and not of biological origin. We have now updated the explanation for why the samples were discarded in the legend of Fig. S1A and thank the reviewer for bringing it to our attention.

2. The authors state that these experiments were performed to better understand the transition steps as cells lose pluripotency and differentiate into various germ layers and that this requires continuous information of an individual cell's state at any time point. However, the experimental design here only studies Nanog reporter expression dynamics, and all other measurements are carried out at an endpoint. Moreover, several studies have previously analyzed Nanog dynamics under different

medium conditions [1-4], and the study here did not generate any new insights based on these data on early ESC lineage specification. Thus, the purpose/utility of Nanog tracking here is unclear.

Thank you. We agree with the reviewer that the results regarding NANOG dynamics confirm previous studies (including our own). NANOG is a key pluripotency marker, and NANOG low cells are known to be more prone to differentiation (Abranches et al., 2014; Chambers et al., 2007; Hastreiter & Schroeder, 2016). However, the studies we have seen that relate NANOG state to differentiation potential typically sort for NANOG low cells in one timepoint, replat them in defined pluripotency media like 2iL (Ying et al., 2008) and count the colonies that emerge to identify pluripotency potential (Abranches et al., 2014; Chambers et al., 2007; Kalkan et al., 2017). To our knowledge, no analysis had been done on NANOG dynamics over cell generations together with how that relates to an end cell state at the high dimensional protein quantification provided here, which is several folds more comprehensive than previous immunostainings, in particular how it relates to lineage commitment. Here, our goal was to identify if any specific dynamics in the ancestors of early lineage committed cells can be used as a predictive tool for lineage commitment later in time. Previous studies (Filipczyk et al., 2015; Hastreiter & Schroeder, 2016) quantifying Nanog dynamics over multiple generations did so only in pluripotency media SerumLIF, and only used 2-3 pluripotency markers to identify possible end states and thus identify a link between the two. The increased number of markers used here including multiple lineage markers and Nanog dynamics in differentiation media could have identified additional previously unrecognized states which could have had different differentiation pathways with different Nanog kinetics. The information that this is not the case, which we provide here, is therefore useful for the community.

In addition, we have now extended this analysis of only NANOG dynamics to also FOXA2, BRACHYURY or any cell fate in the revised manuscript (Supp. Fig. 4).

3. In RA medium, around 1% of cells coexpressed SOX1 and FOXA2. The authors hypothesize that these cells represent bipotential neuroectoderm/definitive endoderm precursors. The current manuscript states that "We identify a novel developmental cell population and find unexpected lineage plasticity..." However, there is no attempt to further examine these cells and thus this statement is not supported. Since this is one of the main conclusions of this manuscript, further characterization of these cells is necessary:

- Specifically, while the authors use FOXA2 as a definitive endoderm marker, it is also expressed in other lineages e.g. extraembryonic primitive endoderm, visceral endoderm (including anterior visceral endoderm), cardiac progenitors [5], floor plate, axial mesoderm, and dopaminergic neurons. The authors should sort the SOX1- FOXA2+, SOX1+ FOXA2-, and SOX1+ FOXA2+ cells and perform in-depth transcriptional characterization to distinguish between these possibilities and particularly to understand what the FOXA2-expressing cells represent. Ideally, this would be sequencing but, at the least, a comprehensive panel of markers that can clearly distinguish between each of these lineages.*
- The authors should sort individual SOX1+FOXA2+ cells and show that they can give rise to both endoderm and neuroectoderm (based on comprehensive marker analysis).*

Thank you, we have now performed these RNASeq experiments to characterize the lineages of these cells based on their transcriptome (Fig. 5, Supp. Fig. 6). Briefly, based on transcriptome signatures and cell morphology, we have identified SOX1-FOXA2+ to be visceral/parietal endoderm cells. They are not floor plate cells, axial mesoderm or cardiac progenitors as these cell types arise from mesoendoderm, marked by expression of Brachyury, Gsc, Eomes etc. which are not expressed in these SOX1-FOXA2+ cells. SOX1+FOXA2+ cells have a unique cell state expressing both neuroectoderm markers and endoderm markers and can give rise to both populations based on morphology and transcriptome signature of the progeny (Fig. 5, Supp. Fig. 6). (Please also note the

timing of SOX1+FOXA2+ cells' first appearance in the embryo preceding notochord and later floor plate formation, see below)

- In Fig. 3F, the authors show scatter plots of NANOG vs. FOXA2 and NANOG vs. SOX1 expression levels. Since the focus of these figures is the FOXA2+ SOX1+ cells, the authors should also show the same type of scatter graph of FOXA2 vs. SOX1 expression so that the double positive population can be clearly visualized in a quantitative manner.

Thank you, we now also provide the additional scatter plot of SOX1 vs FOXA2 expression as requested by reviewer in revised Fig. 3F, allowing clear visualization of SOX1+FOXA2+ population.

- Importantly, no attempt is made to determine whether this SOX1+ FOXA2+ population exists in the embryo in vivo. As such the authors cannot claim to identify a "novel developmental cell population". The authors should examine this either by antibody staining and/or chimera generation using their double reporter cell lines.

We have now performed large-volume SOX1 and FOXA2 immunostaining of every cell in post implantation embryos at various stages of gastrulation. We were indeed able to identify a SOX1+FOXA2+ population also in vivo (revised Fig. 6 and Supp. Fig. 7). SOX1+FOXA2+ cells are first observed in the ventral tip of the neural groove of E7.5 embryos, in direct contact with both SOX1-FOXA2+ visceral endoderm cells and SOX1+FOXA2- neuro ectoderm cells. To our knowledge, this is the first time such a population has been discovered in vivo. At E8.5, we also found a SOX1+FOXA2+ cell population at the ventral tip of the neural tube, which is formed with closing of the neural groove, co-expressing NKX2.2, a marker for p3 neural progenitors(Delás et al., 2023). It has previously been hypothesized that the notochord (of mesodermal origin and expressing FOXA2) induces neural floor plate formation as well as specification of neural tube progenitors via the Shh morphogen (Delás et al., 2023; Lek et al., 2010). Our findings now suggests that this p3 progenitor population has its origin in the SOX1+FOXA2+ cells observed at E7.5 and not from the notochord based on both timing and location. Timing-wise, FOXA2 is expressed in the neural groove at E7.5 prior to the formation of the notochord, with its precursor the node clearly marked by FOXA2+ and in a spatially different part of the embryo. Location-wise, SOX1+FOXA2+ cells at E7.5 and E8.5 and SOX1+FOXA2+ NKX2.2+ cells at E8.5 are expressed in a specific ventral location (revised Fig. 6 and Supp. Fig. 7). In addition, quantification revealed that a majority of NKX2.2+ are also SOX1+FOXA2+, especially in the younger embryos of ~E8.5. This suggests that the observed SOX1+FOXA2+ population in vivo is a product of the interaction between neuroectoderm cells in close contact with visceral endoderm cells, consistent with SOX1+FOXA2+ cells characterized in vitro.

4. The authors generate a SOX1 FOXA2 double reporter cell line. However, the half-life of the specific fluorescent reporters used will confound the identification of a true SOX1+ FOXA2+ population. The authors should discuss this and at least compare the proportion of their cultures that are double positive using their reporter compared to their antibody staining.

Thank you. Indeed, stability of fluorescent reporters can confound the identification of true SOX1+FOXA2+ populations, if the fluorescent marker expression gets decoupled from protein expression. This is typically an issue with a snapshot analysis when the timing of marker expression is not known, and a stable fluorescent protein is used as a proxy for marker expression while the marker might already be off. However, during and soon after the onset of marker expression, it is expected that the fluorescent protein reports marker expression accurately. Applying live imaging to quantify SOX1 and FOXA2 dynamics during lineage commitment (Fig. 4D), we use the fluorescent proteins as a marker for expression onset. In this time window, during marker upregulation, we believe it is reasonable to assume EGFP and mCherry accurately report SOX1 and FOXA2 expression respectively. In addition, we think that the suggested comparison of proportions of cells obtained by different methods to infer accuracy of fluorescent reporters is dangerous. This is because e.g. of

differences in sensitivity of fluorescence detection in immunostaining vs FACS and the method of analyzing imaging data vs FACS data. Therefore, to quantify how well reporter expression identifies protein expression, we had stained for SOX1 and FOXA2 protein expression in the exact same Sox1-EGFP/FOXA2mCherry cells for which we also had GFP and mCherry fluorescence intensity values and found that reporter expression well correlates with SOX1 and FOXA2 protein expression at single cell level at 2 days post RA differentiation (Fig. 4B). We have now extended this quantification to 4- and 6-days post RA differentiation to see if reporter accuracy decreases with time (revised Supp Fig. 5A). We found that FOXA2mCherry expression remains highly correlated to FOXA2 protein expression, corroborated also by RNASeq data where FoxA2 RNA expression is upregulated in FOXA2mCherry+ cells across 6 days of differentiation (Fig. 5D). Sox1-EGFP expression remains highly correlated with SOX1 protein expression also at 4 days post differentiation but becomes less correlated at 6 days post RA differentiation. Since for Sox1-EGFP, the fluorescent reporter is not directly fused to Sox1 gene but replaces one of the Sox1 alleles, it is to be expected that it becomes less accurate with time. This Sox1-EGFP/FoxA2mCherry line was generated from the pre-existing Sox1-EGFP reporter line made in Austin Smith's lab (Aubert et al., 2003) and extensively used by multiple labs to identify cells committed to neuroectoderm lineage (Barraud et al., 2005; Incitti et al., 2014; Lu et al., 2009). For the timespan of our experiments here, we consider the Sox1-EGFP and FOXA2mCherry fluorescent reporters to be a reliable reflection of SOX1 and FOXA2 protein expressions.

Moreover, the authors should perform a flow cytometry time course of their double reporter cell line in RA culture.

Thank you, we have now also conducted a flow cytometry time course of the Sox1-EGFP/FOXA2mCherry line over 6 days of RA differentiation and the results are in Supp. Fig. 5C-5D. The percentage of Sox1+FOXA2- cells peaks at 2 days post differentiation before gradually returning to baseline by D6, consistent with literature (Lu et al., 2009). In contrast, the proportion of Sox1-FOXA2+ cells increases with time, while the proportion of Sox1+FOXA2+ cells remains broadly consistent over time.

5. In Fig. 2, the authors compare the Nanog expression state of 'Decision cells'. The conclusion is that 1) most decision cells are Nanog-, and 2) the % of Nanog+ cells is higher in earlier Decision cell ancestral generations. This is not surprising since most cells in RA cultures as a whole are Nanog- and the proportion of Nanog+ cells decreases over time, therefore earlier generations will have fewer Nanog- cells across the entire culture. Although the authors compare the proportions of Nanog+ and Nanog- cells in Sox1+ vs Sox1- cells and their ancestors, they do not say whether the differences are statistically significant. If they are not statistically significant, this data is not informative regarding neuroectoderm specification. Another caveat to this analysis is that based on the nomenclature of "Decision cells", it seems the authors consider Sox1- cells as undifferentiated or not having made a "lineage decision". At least a fraction of the Sox1- cells may also have differentiated but toward a non-neuroectoderm lineage and hence Nanog dynamics may be similar between Sox1- and Sox1+ differentiating cells.

Thank you. We have now conducted statistical tests to identify if the proportion of cells having NANOG- dynamics over generations are significantly different between SOX1+ decision cells and SOX1- cells. The statistical test supports our observations that NANOG- proportions are non-significant when considering lineage data and NANOG reporter expression from 2 generations and above. For Decision Cells and Decision Cells + 1 generation up, the proportions of NANOG- dynamics are significantly different compared to Sox1- cells. However, the effect size is too small to have predictive power in identifying SOX1+ lineage commitment with difference in NANOG- proportions at Current Generation and Current Generation + 1 generation up between SOX1+ decision cells and SOX1- cells (at 10% and 13%, respectively). These results are now in revised Fig. 2C and are expanded in text as well.

Further, keeping in mind that some of the SOX1- cells may have differentiated to another lineage as the reviewer pointed out, we repeated the 'Decision Cell' analysis considering any fate acquisition (neuroectoderm or non-neuroectoderm) to be equivalent to identify if NANOG reporter dynamics are different when all cells that are lineage committed are grouped together compared to the remaining cells that have not upregulated any lineage marker (Supp Fig. 4B). This did not alter our conclusions. While proportions of NANOG- dynamics are significantly different at Decision Cell and Decision Cell + 1 generation up, the effect size again is too small to have predictive power in identifying lineage commitment. At Decision Cell + 2 generations up and onwards, the differences in proportions are not significant.

Additional questions and concerns:

1. Fig 1: the fluorescence level of the Nanog reporter increases around the time of cell division. Is this an artefact of the change in cell morphology at this time and, if so, is there some normalization that is needed at the time of cell division? The authors should discuss this.

Indeed, around cell division the morphology of the cells alters due to mitosis leading to artefact signals. To counter this, for any analysis involving NANOG reporter signal, we excluded the signal from the two timepoints before and after cell division(as was described in the Methods section under "Actual NANOG transition dynamics in SerumLIF"). We have now also stated this explicitly in the results part of Fig.2.

2. The authors indicate that the probability of a cell transitioning from a Nanog- to a Nanog+ state is much higher than the other way around. This is surprising since Nanog is a marker of the pluripotent state and therefore it might be expected that it is more common for Nanog expression to be lost (i.e. through spontaneous differentiation) than gained. The authors should discuss this observation.

The transition probabilities were calculated in pluripotency supporting SerumLIF medium, where cells can be maintained in a stable pluripotent state, potentially indefinitely. In such a scenario, even if Nanog is downregulated in one generation, the surrounding network of pluripotent proteins and the media conditions themselves, likely push the cell back into a NANOG+ state as it is a more stable state in this medium. This is consistent with literature that shows NANOG- state in SerumLIF is reversible(Filipczyk et al., 2015). The presence of a persistent, minor NANOG- negative population in SerumLIF, that nevertheless, doesn't increase in proportion with increasing generations (Fig. 2A, Supp. Fig. 2C) is also indicative of the unstable nature of a NANOG- state in pluripotent media. We have now explained in the text our reasoning for why the probabilities are not unexpected.

3. In Fig. 2D, the meaning of "cell lifetime" is unclear. Does this refer to actual cell lifetime i.e. time before cell death or time before cell division, or a combination of both of these? This should be stated more explicitly in the text.

It refers to the time from birth to death or division of a cell. In Fig. 2D, we removed all cells from Generation 0 and all cells from last generation (i.e.: Last Nodes of a tree), since the complete Cell lifetime is not known for these cells. Only cells were used for analysis that originated from a cell division and ended in a cell division during the observation period. We have now stated this more explicitly also in the text in the Results part.

4. In Fig. 4C, the authors should also provide overlay images.

Thank you, we tried different pseudo colors, but it is difficult to make accurate overlay pictures where 4 channels are overlapping and over roughly same location. We feel that in such cases, gray scale images give better indication of the signal. Especially since the intensity in overlay images is dependent on the contrast we have set manually and can be misleading. Therefore, we have now placed arrows on the images to better guide the reader on which cells are supposed to be the same

across as images. In addition, we have provided overlay images for 2 channels each – Sox1-EGFP/FoxA2mCherry and Sox1 antibody/FoxA2 antibody in Supp Fig. 5B as an additional way of representing the images in Fig. 4C.

5. The Discussion section states that since there was no difference in the 37 markers analyzed between SOX1+ and SOX1- sisters that there may be no transition steps between loss of pluripotency and neuroectoderm differentiation. Previous studies of early ESC differentiation have already demonstrated a dismantling of the pluripotency network within the first 25 hrs of differentiation in a medium permissive for neural specification [6] and thus this speculation is incorrect and should be removed.

Thank you, we have removed this sentence.

We would also like to point two additional corrections we made in the revised manuscript:

- In the previous version, in Fig. 1B, a wrong example image for Oct6 was used. This has now been corrected in the revised manuscript.

- In the previous version, in Fig. 3C, the n numbers in the legend read 'n> 12000 cells'. It should have been 'n> 4000 NANOG- cells'. We have corrected this in the revised manuscript.

References

- Abranches, E., Guedes, A. M. V, Moravec, M., Maamar, H., Svoboda, P., Raj, A., & Henrique, D. (2014). Stochastic NANOG fluctuations allow mouse embryonic stem cells to explore pluripotency. *Development (Cambridge)*, *141*(14), 2770–2779. <https://doi.org/10.1242/dev.108910>
- Acampora, D., Di Giovannantonio, L. G., & Simeone, A. (2013). Otx2 is an intrinsic determinant of the embryonic stem cell state and is required for transition to a stable epiblast stem cell condition. *Development*, *140*(1), 43–55. <https://doi.org/10.1242/dev.085290>
- Aubert, J., Stavridis, M. P., Tweedie, S., O'Reilly, M., Vierlinger, K., Li, M., Ghazal, P., Pratt, T., Mason, J. O., Roy, D., & Smith, A. (2003). Screening for mammalian neural genes via fluorescence-activated cell sorter purification of neural precursors from Sox1-gfp knock-in mice. *Proceedings of the National Academy of Sciences*, *100*(suppl 1), 11836–11841. <https://doi.org/10.1073/PNAS.1734197100>
- Barraud, P., Thompson, L., Kirik, D., Björklund, A., & Parmar, M. (2005). Isolation and characterization of neural precursor cells from the Sox1 –GFP reporter mouse. *European Journal of Neuroscience*, *22*(7), 1555–1569. <https://doi.org/10.1111/j.1460-9568.2005.04352.x>
- Chambers, I., Silva, J., Colby, D., Nichols, J., Nijmeijer, B., Robertson, M., Vrana, J., Jones, K., Grotewold, L., & Smith, A. (2007). Nanog safeguards pluripotency and mediates germline development. *Nature*, *450*(7173), 1230–1234. <https://doi.org/10.1038/nature06403>
- Delás, M. J., Kalaitzis, C. M., Fawzi, T., Demuth, M., Zhang, I., Stuart, H. T., Costantini, E., Ivanovitch, K., Tanaka, E. M., & Briscoe, J. (2023). Developmental cell fate choice in neural tube progenitors employs two distinct cis-regulatory strategies. *Developmental Cell*, *58*(1), 3-17.e8. <https://doi.org/10.1016/j.devcel.2022.11.016>
- Filipczyk, A., Marr, C., Hastreiter, S., Feigelman, J., Schwarzfischer, M., Hoppe, P. S., Loeffler, D., Kokkaliaris, K. D., Endele, M., Schauburger, B., Hilsenbeck, O., Skylaki, S., Hasenauer, J.,

- Anastassiadis, K., Theis, F. J., & Schroeder, T. (2015). Network plasticity of pluripotency transcription factors in embryonic stem cells. *Nature Cell Biology*, *17*(10), 1235–1246. <https://doi.org/10.1038/ncb3237>
- Hastreiter, S., & Schroeder, T. (2016). Nanog dynamics in single embryonic stem cells. *Cell Cycle*, *15*(6), 770–771. <https://doi.org/10.1080/15384101.2015.1137711>
- Heezen, L. G. M., Abdelaal, T., van Putten, M., Aartsma-Rus, A., Mahfouz, A., & Spitali, P. (2023). Spatial transcriptomics reveal markers of histopathological changes in Duchenne muscular dystrophy mouse models. *Nature Communications*, *14*(1), 4909. <https://doi.org/10.1038/s41467-023-40555-9>
- Incitti, T., Messina, A., Bozzi, Y., & Casarosa, S. (2014). Sorting of Sox1-GFP Mouse Embryonic Stem Cells Enhances Neuronal Identity Acquisition upon Factor-Free Monolayer Differentiation. *BioResearch Open Access*, *3*(3), 127–135. <https://doi.org/10.1089/biores.2014.0009>
- Kalkan, T., Olova, N., Roode, M., Mulas, C., Lee, H. J., Nett, I., Marks, H., Walker, R., Stunnenberg, H. G., Lilley, K. S., Nichols, J., Reik, W., Bertone, P., & Smith, A. (2017). Tracking the embryonic stem cell transition from ground state pluripotency. *Development (Cambridge)*, *144*(7), 1221–1234. <https://doi.org/10.1242/dev.142711>
- Lek, M., Dias, J. M., Marklund, U., Uhde, C. W., Kurdija, S., Lei, Q., Sussel, L., Rubenstein, J. L., Matise, M. P., Arnold, H.-H., Jessell, T. M., & Ericson, J. (2010). A homeodomain feedback circuit underlies step-function interpretation of a Shh morphogen gradient during ventral neural patterning. *Development*, *137*(23), 4051–4060. <https://doi.org/10.1242/dev.054288>
- Lu, J., Tan, L., Li, P., Gao, H., Fang, B., Ye, S., Geng, Z., Zheng, P., & Song, H. (2009). All-trans retinoic acid promotes neural lineage entry by pluripotent embryonic stem cells via multiple pathways. *BMC Cell Biology*, *10*. <https://doi.org/10.1186/1471-2121-10-57>
- Múnera, J. O., Kechele, D. O., Bouffi, C., Qu, N., Jing, R., Maity, P., Enriquez, J. R., Han, L., Campbell, I., Mahe, M. M., McCauley, H. A., Zhang, X., Sundaram, N., Hudson, J. R., Zarsozo-Lacoste, A., Pradhan, S., Tominaga, K., Sanchez, J. G., Weiss, A. A., ... Wells, J. M. (2023). Development of functional resident macrophages in human pluripotent stem cell-derived colonic organoids and human fetal colon. *Cell Stem Cell*, *30*(11), 1434-1451.e9. <https://doi.org/10.1016/j.stem.2023.10.002>
- Takada, H., Sasagawa, Y., Yoshimura, M., Tanaka, K., Iwayama, Y., Hayashi, T., Isomura-Matoba, A., Nikaido, I., & Kurisaki, A. (2023). Single-cell transcriptomics uncovers EGFR signaling-mediated gastric progenitor cell differentiation in stomach homeostasis. *Nature Communications*, *14*(1), 3750. <https://doi.org/10.1038/s41467-023-39113-0>
- Ying, Q. L., Wray, J., Nichols, J., Batlle-Morera, L., Doble, B., Woodgett, J., Cohen, P., & Smith, A. (2008). The ground state of embryonic stem cell self-renewal. *Nature*, *453*(7194), 519–523. <https://doi.org/10.1038/nature06968>

Reviewer references:

1. Abranches, E., et al., Stochastic NANOG fluctuations allow mouse embryonic stem cells to explore pluripotency. *Development*, 2014. *141*(14): p. 2770-9.

2. Pezzarossa, A., et al., *Imaging Pluripotency: Time-Lapse Analysis of Mouse Embryonic Stem Cells. Methods Mol Biol*, 2016. 1341: p. 87-100.
3. Hastreiter, S., et al., *Inductive and Selective Effects of GSK3 and MEK Inhibition on Nanog Heterogeneity in Embryonic Stem Cells. Stem Cell Reports*, 2018. 11(1): p. 58-69.
4. Jin, K.X., et al., *N6-methyladenosine (m(6)A) depletion regulates pluripotency exit by activating signaling pathways in embryonic stem cells. Proceedings of the National Academy of Sciences of the United States of America*, 2021. 118(51).
5. Bardot, E., et al., *Foxa2 identifies a cardiac progenitor population with ventricular differentiation potential. Nature Communications*, 2017. 8.
6. Kalkan, T., et al., *Tracking the embryonic stem cell transition from ground state pluripotency. Development*, 2017. 144(7): p. 1221-1234.

Reviewer #2 (Remarks to the Author):

The manuscript presented by Arekatla et al. combines cell tracking with high-dimensional protein profiling by Imaging Mass Cytometry (IMC) to simultaneously study the dynamics of ESCs differentiation and characterize their molecular features at the protein level. The approach employed is interesting and is within the continuing trend of connecting lineages, cell fate choice and single-cell transcriptomics/proteomics. The authors first perform time lapse microscopy to lineages and NANOG dynamics across 5-6 generations during 46 h differentiation of ESCs and IMC at the end to collect 37 proteins data. Then they evaluate NANOG expression and dynamic patterns and correlate its behavior with SOX1 (neuroectoderm lineage marker) expression patterns. They observe that NANOG is downregulated within 2-3 generations but is not required to for neuroectoderm cell fate choice. Then they found cells co-expressing SOX1 and FOXA2 (definitive endoderm marker), generate double reporter cell line to evaluate their expression dynamics during neuroectoderm differentiation. While the approach is interesting, the data and conclusions suffer from short differentiation time and data is underutilized. The observation of SOX1 and FOXA2 cells is interesting but validation of their molecular nature and functional potential is not resolved. It would have been useful if authors substantiated solid evidence to the set of observations presented, towards gaining new significant insights.

Thank you for your time and feedback!

1) The differentiation time used (46 h) in the study is rather quite short to establish exclusive germ layer cell fates. While SOX1 expression can be observed by 46 h, its expression level as well as the proportion of cells are meaningfully distinct at later time points (PMID: 12524553). As a result, often 72 – 96 h is often used to differentiate ESCs to germ layer precursors. The following in the data make this reviewer suspect whether the differentiation time used is optimal.

A) In Fig 1C, SOX1 levels is relatively high in cells of SerumLIF (SL) than in RA, the condition used to induce neuroectoderm differentiation.

B) Similarly in Fig S1B, Brachyury levels in SerumLIF (SL) is relatively same as in Meso, the condition used to induce mesendoderm differentiation.

C) In Fig 1D with dimensionality reduction, distinct differentiating subpopulations did not emerge. That the majority of pluripotency transcription factors and other regulators measured by IMC are still highly expressed suggests that there may not have been sufficient time for cells to segregate into distinct germ layer lineages.

D) The potentially interesting role of FOXA2+ and FOXA2+ SOX1+ cells remain unresolved.

Thank you. The idea behind this study was to capture the intermediate transition states and the very first steps of pluripotency exit to lineage commitment. That is, fewer cells that are already committed to a germ layer lineage and more cells in intermediate transition states, if any. One reason we did not go for a longer differentiation time is that while it might lead to more cells segregating into different lineages, the potentially interesting early transition states from pluripotency, where decisions are likely made, would be lost as more cells differentiate and mature cell types with distinct morphology and separated clusters emerge, long after the decision making. Together with the massively increasing complexity of cell segmentation and tracking at later time points, we therefore chose 46 h as an optimum time. There indeed was an exploratory aspect to this study since no previous study has been conducted combining lineage data with comprehensive protein data to identify first steps to germ layer fate commitment from ESCs to understand if ancestral information of a cell could predict future fate. In this process, we discovered the

SOX1+FOXA2+ cells as well as the FOXA2+ cells, that were interesting as the reviewer points out and interrogated the dynamics of these transcription factors further. Based on this and other reviewers' comments, we have now further characterized these cells by RNASeq and stained for these markers also in the post-implantation gastrulating embryo to confirm that this population can also be found in vivo. The results of these new experiments are in Fig. 5-6 and Supp. Fig. 6-7.

2) Majority of the cells in IMC image (Fig 1B) appear to be expressing SOX1. It is surprising given the 46 h differentiation time and that authors themselves found about 9% of SOX1+ cells (Fig 3C). A clear comparison of lineage markers signal observed in IMC, antibody staining and live-cell reporter is required to demonstrate the specificity of the antibody-isotope conjugates. It is understood that SOX1 levels are thresholded as in Fig 1E for subsequent analysis. However, it is key to show that the IMC reagents for lineage markers truly reflect their cell fate specific expression.

Thank you. We had extensively tested the suitability of the antibodies prior to metal conjugation. In the case of SOX1, most cells being positive is only true for the one colony shown in the example image in Fig. 1B. This is one colony, a small zoomed in portion from one Field of View, of > 10 Fields of View per experimental condition. When quantifying the complete Fields of View, the majority of cells are SOX1- and that is consistent with the proportions observed by us. We have included a larger imaging area in revised Fig. S1B to better illustrate SOX1 expression.

A further proof that our IMC antibodies detect specific signal is: 1) Independent thresholding on NanogVenus reporter and NANOG IMC expression giving similar NANOG+ and NANOG- proportions (Fig. 1F, 2A, Supp Fig. 3C). 2) Once we detected the rare SOX1+FOXA2+ population, we rechecked and confirmed that such a population indeed exists via immunostaining in different ESC lines, thus further validating that the lineage marker expression observed by us in IMC was specific. This was further confirmed by Sox1-EGFP/FOXA2mcherry fluorescent reporter expression. In general, with imaging data it is always key to examine the raw images behind the quantifications. All antibodies were selected because they had sufficient dynamic range such that upregulation could be clearly observed by eye, and it is important to get a feel for what true upregulation looks like compared to background noise by examining the raw images. Please see also response to your (this reviewer's) question 4 where we provide further complete Fields of View of SOX1 staining.

3) Even though this represents an interesting data set, a total of 37 protein panel consists of multiple lineage markers (SOX1, BRACHYURY, GATA6, OTX2) and two different differentiation conditions, the data remains underutilized with most analysis restricted to SOX1 expression patterns. How the other markers and their associated lineage patterns vary is not evaluated. Within the observed SOX1 patterns, it is also not clear how the patterns of all proteins vary and whether they can explain the cell state phenotypes.

Thank you. We have now repeated the analysis performed with SOX1 decision cells with other lineage markers such as BRACHYURY, FOXA2 and a combination of all lineage markers grouped as one. We did not include OTX2 in this analysis as it is not reported to be a lineage marker but is expressed also in pluripotent cells (Acampora et al., 2013) and upregulated in many cells in general during RA differentiation. Additional analysis was consistent with results obtained for SOX1 lineage commitment in that ancestral Nanog dynamics, cell lifetime or motility do not predict lineage commitment for any of the lineages analyzed.

Further, within SOX1 patterns, we had already included what the reviewer suggested in Fig. 3A, where we analyzed 28 sister pairs where one cell was SOX1+ and other SOX1- for which full IMC data was available. Expression patterns of all proteins in the IMC panel were analyzed between the two sisters to identify if other protein expressions also vary with lineage commitment and could explain SOX1+ lineage commitment decision, but we did not find this to be the case (Fig. 3A). We also now added this analysis for FOXA2+ and FOXA2- sisters as well Any Germ Layer Fate+ vs Germ Layer Fate-

cells to identify potential regulators of lineage commitment but no difference in expression of other proteins was observed. The additional analysis is in Supp Fig. 4.

4) *Unbiased choice of thresholds based on gamma distribution fit is interesting and appears to be robust. Did the authors test similar unbiased thresholds for lineage markers such as SOX1, instead of the current choice from what appears to be more subjective? Also, if we go by the expression profile of SOX1 in Fig 1C, in contrast to results in Fig 1E is it likely that there would be more SOX1+ cells in SL than in RA?*

Thank you. Gamma distribution fits are a good choice where there is a clear bimodal population. For antibodies like SOX1, where there is a slight shift to right in distribution profile in SerumLIF compared to RA and only a minority of cells are SOX1+, the values on the x-axis also must be considered and if a shift of ~ 0.1 is biologically meaningful. The dynamic range of any antibody depends on the quality of the antibody, the metal isotope it is tagged with and the cell population under assay. Therefore, it is very important to do manual checks for each antibody on the primary imaging data to get a feel for what true signal looks like vs background noise following image processing and conversion of imaging pixel data into protein intensities. We have now included more primary IMC data in Supp Fig. 1B to give readers a better idea of how the signal looks like across a wide range of cells, not just in one sample colony as shown previously in Fig. 1B. Further, please see below an example of how SOX1 signal looks in SerumLIF vs RA. All image processing is kept constant between both images. Scale bar 50 μm . True signal appears bright and concentrated as in some cells in RA while in SerumLIF, this signal is low, speckled, diffuse and thus classified to be background.

5) *“Nanog downregulation...is not sufficient for SOX1+ lineage commitment” (line 147). It is not surprising and is an expected result. NANOG is known to be downregulated at faster rate than other pluripotency factors during ESCs differentiation, resulting in dismantling of pluripotency network. While NANOG low cells are in general more prone to differentiation, it is not expected to suppress or promote germ layer precursor cell fates. However, NANOG (along with OCT4) is known to promote further downstream cell fates such as definitive endoderm (PMID: 21245162). Longer time differentiation analysis combined with perturbation tests would have revealed new insights for NANOG’s role in differentiation.*

Thank you. We agree with the reviewer that our results regarding NANOG dynamics confirm

expectations from the literature derived from other approaches. NANOG is a key pluripotency marker and NANOG low cells are known to be more prone to differentiation (Abranches et al., 2014; Chambers et al., 2007; Hastreiter & Schroeder, 2016). However, the studies we have seen that relate NANOG state to differentiation potential typically sort for NANOG low cells at one timepoint, replate them in defined pluripotency media like 2iL (Ying et al., 2008) and count the colonies that emerge to identify pluripotency potential (Abranches et al., 2014; Chambers et al., 2007; Kalkan et al., 2017). To our knowledge, no analysis had been done on NANOG dynamics over generations and how that relates to an expression end cell state with information about multiple lineage marker proteins and utilizing kinship information. Previous studies (Filipczyk et al., 2015; Hastreiter & Schroeder, 2016) quantifying Nanog dynamics over multiple generations did so only in pluripotency media SerumLIF and used 2-3 pluripotency markers to identify possible end states and thus identify a link between the two. Here, our goal was to identify if any specific dynamics in the ancestors of early lineage committed cells can be used as a predictive tool for later lineage commitment. While we do not find clear predictors, this was not obvious before actually quantifying it, and is therefore useful information for the field. We have also extended this analysis on NANOG dynamics to FOXA2, BRACHYURY or any cell fate now (Supp. Fig. 4) as suggested by this reviewer in Point 3. We feel that additional analyses, e.g. whether NANOG dynamics might play an interesting role in more mature differentiation steps, would go beyond the scope of this paper, which already has quite a lot of diverse points and data. In addition, perturbation tests alter the natural cell states of cells, and dynamic information is lost when we e.g., conditionally knock-out a gene, whereas our experimental strategy is to analyze the natural dynamics of molecular regulators to identify how different cell fates emerge without experimental intervention.

6) Observation of the unexpected SOX1+ FOXA2+ cells is interesting. However, the evidence provided in the paper does not resolve whether they represent “novel” differentiation stage as precursors for neuro-ectoderm and definitive endoderm lineages. The evidence provided is mostly observational and solid evidence is required to validate the hypothesis. For instance: i) verifying FOXA2 expressing cells with other definitive endoderm markers; ii) perturbation tests to validate the novel differentiation stage model; iii) long-term differentiation analysis in combination with lineage tracing, IMC and perturbation tests; iv) clear demonstration of their functional differentiation ability (at least in-vitro). Thank you, we have now performed RNASeq experiments to characterize the lineages of these cells based on their transcriptome. In addition, we performed SOX1 and FOXA2 antibody staining in post implantation embryos at various stages of gastrulation to demonstrate that a SOX1+FOXA2+ population is indeed found in-vivo at specific developmental stages and locations. Our results are now presented in Fig. 5-6 and Supp. Fig. 6-7. Briefly, based on transcriptome signatures and cell morphology, we have identified SOX1-FOXA2+ to be visceral/parietal endoderm cells. SOX1+FOXA2+ cells have a unique cell state expressing both neuroectoderm markers and endoderm markers and can give rise to both populations based on morphology and transcriptome signature of the progeny. Furthermore, antibody stainings of embryos revealed the presence of SOX1+FOXA2+ cells, first observed in the ventral tip of the neural groove of E7.5 embryos, in direct contact with both SOX1-FOXA2+ visceral endoderm cells and SOX1+FOXA2- neuro ectoderm cells. We did not do perturbation tests as we are unsure how disrupting the cell state of this novel population would help in further characterization of the cells. As we have SOX1+FOXA2- and SOX1-FOXA2+ cells occurring biologically without external manipulation that serve as natural controls against which to compare the state of these cells and their progeny.

Other minor points:

7) *Is the differentiation time 48 h or 46 h? In Fig 1A it is 48 h but the in rest of the text it is 46 h.*
Thank you! It should be 46h also in Fig. 1A as it is in rest of the text where experiments with end-point IMC were done. We have now corrected it.

8) *Is the panel of 37 antibody-isotope reagents custom generated by manual conjugation? If so how do these reagents act as novel resource compared to the existing CyTOF reagents to study ESC regulatory factors?*

Reagents to study ESC regulatory factors have only recently been validated only for suspension, not for imaging CyTOF (Meharwade et al, Methods 2022). Our custom panel, generated by manual conjugation, has been optimized for Imaging Mass Cytometry. The combination of live cell imaging and 37-dimensional protein cell state of the same cell are only possible with Imaging Mass Cytometry. In addition, in situ measurements like Imaging Mass Cytometry also avoid changes in cell state associated with trypsinization and dissociation necessary for suspension methods.

9) *Which isotope is conjugated to which antibody? Specific list of isotopes used should be listed as part of Supplementary table 1.*

Thank you, we have now also added the isotopes used to Supplementary table 1.

10) *In Fig S1B, what does 'phPHOSPHO' and 'TFE3' represent? TFE3 is not in Table 1 and pSMAD1/5 distribution profile is not in Fig S1B.*

Thank you, phPHOSPHO is pSMAD1/5. And we have now changed the name accordingly in the distribution profile in Fig. S1B. We have also added information about TFE3 in Supplementary Table 1.

We would also like to point two additional corrections we made in the revised manuscript:

- In the previous version, in Fig. 1B, a wrong example image for Oct6 was used. This has now been corrected in the revised manuscript.

- In the previous version, in Fig. 3C, the n numbers in the legend read 'n> 12000 cells'. It should have been 'n> 4000 NANOG- cells'. We have corrected this in the revised manuscript.

Reviewer #3 (Remarks to the Author):

Overview: Authors developed an approach for live cell fate tracking with an end-point Imaging Mass Cytometry measurement. This approach allows to track to create cell pedigrees and study end-point cell fate distribution among kin cells, as well as to correlate it with cell's life cycle and motility. This approach also allows to calculate cell fate transition probabilities informed by fluorescence of a live reporter (in this case fused to Nanog protein). Authors also engineered a reporter cell line expressing fluorescent SOX1 and FOXA2, which allowed them to show mutual dynamics of these transcriptional factors.

The three main findings of the paper in my opinion are:

- SOX1 expression is preceded by Nanog downregulation in the cell's pedigree; however, Nanog downregulation alone is not sufficient for SOX1 cell fate,
- SOX1+FOXA+ cells are not an intermediate for SOX1+ or FOXA+ cell fates, these states exist independently,
- SOX1 expression always precedes SOX1+FOXA+ cell fate and never vice versa.

Using the platform established in this paper opens endless possibilities to study other known transcription factor dynamics in various directed differentiation protocols.

Thank you for your time and feedback!

Minor suggestions/questions:

1. Fig 1B - clarify in the legend that you marked the example cell and its progeny with red circles.

Thank you, we have now clarified this in figure caption of Fig. 1B.

2. Fig 1C - it took me a moment to realize it's a histogram (a distribution?) of values for each protein. May be clarify that in the figure description.

Thank you, we have now clarified in Fig. 1C description that it is a distribution of values for each protein.

3. Did you try to cluster the cells by the "combinatorial thresholded protein expression state"? There can be 2^{37} total cell fates based on binary thresholding for each of the 37 proteins in IMC, but may be some cell fates are preferred? This could reveal a variety of preferred cell fates.

Thank you, revised Supplementary Figure 3B (previously Supplementary Figure 2B) shows exactly this where we identify the top five preferred cell states by condition based on binary classification of protein expression. Here, we only used the proteins for which we could assign meaningful thresholds. For signaling proteins, meaningful thresholds could not be assigned as distribution profiles between conditions practically overlapped with no clear separation. Therefore, for these proteins we utilized the data where we analyzed actual values, e.g.: Umapi (Fig. 1D, Supp Fig. 1C) or where the molecular differences between sister cells with divergent fates were examined (Fig. 3A, Supp. Fig. 4H-I). We have now included in source data the complete list of all cell states observed based on binary thresholding per condition, and their proportions in that condition.

4. Figure 1D: does excluding pHistone and a couple of other "outlier" proteins from analysis improve things?

We presume that by outlier proteins, the reviewer means proteins that indicate cell cycle state or

apoptotic state based on which some distinct clusters emerged. We removed pHISTONE H3, phRB, KI67, CYCLIN B1 and CLEAVED CASPASE from the parameters list for generating umap and ran the analysis again (umap below). As expected, this caused the distinct clusters to merge/move closer to the main big cluster. But otherwise, the shape of Umap remained much the same (see also Fig. 1D). In general, unless there are known issues of antibody signal/quality, we would not recommend removing any antibody from such analysis as we feel more information on a cell state is always better where possible.

5. Line 117: Are the transition probabilities different for different media conditions?

Thank you. Indeed, the transition probabilities are different for different media conditions. In pluripotency promoting SerumLIF media, as stated in manuscript, the probability of NANOG+ cell transitioning to NANOG- cell in the next generation is very low (0.05 ± 0.01) compared to a NANOG- cell transitioning to NANOG+ cell (0.3 ± 0.07). In RA differentiation media, this trend is reversed with the probability of NANOG+ cell transitioning to NANOG- cell in the next generation being higher (0.26 ± 0.09) compared to a NANOG- cell transitioning to NANOG+ cell (0.1 ± 0.07). In Meso differentiation medium, the transition probabilities are roughly similar with NANOG- to NANOG+ being 0.16 ± 0.04 and NANOG- to NANOG+ being 0.12 ± 0.05 . This is likely because of the presence of CHIR99021 inhibitor in Meso differentiation medium, which is also a key component of the pluripotency promoting defined media 2iLIF. We have now included this in Results under section 'NANOG downregulation occurs two generations prior to but is not sufficient for neuroectoderm lineage commitment'.

6. Line 118: why is it obvious that NANOG+ to NANOG- probability is lower than vice versa? NANOG is a pluripotency marker - why would a NANOG- transition to become NANOG+, thus "gaining" pluripotency? Usually, cells tend to lose pluripotency during prolonged culture.

The transition probabilities were calculated in pluripotency supporting SerumLIF medium, where cells can be maintained in a stable pluripotent state, potentially indefinitely. In such a scenario, even if Nanog is downregulated in one generation, the surrounding network of pluripotent proteins and the media conditions themselves, likely push the cell back into a NANOG+ state as it is a more stable state in this medium. This is consistent with literature that shows that the NANOG- state in SerumLIF is reversible (Filipczyk et al., 2015). The presence of a persistent, minor NANOG- negative population in SerumLIF, that doesn't increase in proportion with increasing generations (Fig. 2A, Supp. Fig. 2C) is also indicative of the unstable nature of a NANOG- state in pluripotent media. We have now included the transition probabilities of NANOG in other media in the manuscript and also clarified more

explicitly in that sentence that these probabilities are for SerumLIF. Further, we have also explained in the text our reasoning for why the probabilities are not unexpected.

7. Fig 2 A, B - why does generation 0 in RA condition already have skewed population towards low Nanog? When was the timepoint for generation 0?

Generation 0 denotes all timepoints for which a particular cell was in that generation (until the cells' first division) and all NANOG measurements in those timepoints are plotted for that generation. Therefore, this number varies from cell to cell. The mean timepoint for RA in Generation 0 is 10 which corresponds to ~4hrs in RA media. Keeping also in mind the lag between media exchange and the start of movie acquisition under microscope which can be an additional 1 h, we think this is sufficient time for the population to already start skewing NANOG-.

8. Line 140: could you calculate some metric to prove this statement more stringently? For example, compare raw NANOG levels on the -2 generation level for SOX1+ and SOX1- decision cells and do a t-test and show p-value here? This would greatly strengthen this statement.

Thank you for the suggestion. We have now conducted Fisher's Exact Test to compare the proportion of NANOG dynamics indicated by yellow for each SOX1+ Decision Cell + n Generations to its counterpart in the SOX1- column (revised Fig. 2C) and have now included p values in the revised Fig. 2C, thus testing our observations more stringently by statistical means. Indeed, consistent with our observations, the observed proportions of NANOG- dynamics are significantly different for SOX1+ Decision Cells and Decision Cells + 1 generation up in comparison to proportions of NANOG- dynamics of SOX1- and SOX1- + 1 generation up. But this difference becomes non-significant from 2 generations up and higher. We decided against conducting t-test between raw Nanog values at the -2 generation level because we felt this did not capture the essence of the data that we were presenting. Since the data plotted has lineage information inherent in it and how NANOG dynamics are linked across generations is the key point we wanted to communicate. This information would be lost if we compared only the raw NANOG values per generation. But nevertheless, the reviewer's question pointed us in the right direction, and we thank them for the feedback.

9. Fig 2G - is it possible to zoom in on the violin plots along the y-axis? You can barely see any difference in their means at this scale.

Thank you, we have now put a zoomed in version of Fig. 2G and put the original plot in Supp. Fig 3E.

10. Line 153: how do you define cell lifetime for the last generation cells, where the endpoint IMC measurement was made? What if cells would proceed to live longer if the endpoint experiment was not done? This makes the lifetime measurements inaccurate in that case.

For this analysis, for SOX1+, we identified all cells within a SOX1+ Decision Cell sub-branch for which complete lifetimes were available (cell division to cell division) and excluded all cells of the last generation (i.e. Last nodes) within these sub-branches based on the logic, as the reviewer rightly pointed out, that these cells in last generation might have lived longer prior to fixation. In addition, we excluded all cells in Generation 0, i.e. the cells which were present at the start of experiment and from where we started tracking, because, again, we do not know the full lifetime of these cells. All sub-branches that are not SOX1+ were denoted SOX1- and then, all cells excluded from this data also for which complete lifetimes were not available (last nodes, Generation 0). We have now stated more explicitly in Results how this was calculated, thank you.

11. Line 207: It is hard to see the light purple color (SOX1+FOXA2+ cells) in the figure 3G, since it is only 1% of the population. Therefore, not sure if referencing the Fig 3G for proof of SOX1+FOXA2+ presence makes sense. Either change the representation on Fig 3G or don't reference it here.

Thank you, we have now removed that reference.

12. Where do the negative values in Fig 4B come from? If the plots are in log scale it needs to be indicated in the legend.

These are indeed on the log scale, and we have now indicated this in the legend, thank you.

13. Line 273: how do you explain FOXA2 expression in your experiment then, since no one before observed it in similar experiments?

Our understanding is that no one assayed for it to begin with, and not at such an early time point.

We also only caught it because we were using a 37-antibody panel. If we were staining for 3-4 markers, as is common with normal immunostaining, we would also likely not have included FOXA2 marker in our panel. We have now included this in our discussion.

We would also like to point two additional corrections we made in the revised manuscript:

- In the previous version, in Fig. 1B, a wrong example image for Oct6 was used. This has now been corrected in the revised manuscript.

- In the previous version, in Fig. 3C, the n numbers in the legend read 'n> 12000 cells'. It should have been 'n> 4000 NANOG- cells'. We have corrected this in the revised manuscript.

REVIEWERS' COMMENTS

Reviewer #1 (Remarks to the Author):

The authors have presented compelling new data demonstrating the existence of a SOX2+ FOXA2+ population in the embryo in vivo, with RNA sequencing providing a more in depth characterization of these cells. This significantly enhances the impact of their findings.

However, there are still 2 major points that should be addressed. Most importantly, the authors still do not provide evidence that this population functions as a bi-potent progenitor. Establishing such a definition requires single-cell functional experiments and lineage tracing, neither of which are included in this study. Consequently, any references to these cells as bi-potent progenitors should be omitted, although the authors may speculate on this possibility in the discussion.

Second, the authors frequently describe cells as "lineage committed" or "uncommitted" without supporting functional experiments. Functional assays are necessary to determine a cell's commitment to a specific lineage. While the manuscript classifies cells as "uncommitted" based on a limited set of lineage markers, it is possible that these cells express markers of other lineages not examined, making it inaccurate to label them as "uncommitted." This should also be rectified in the text with more appropriate terminology.

Reviewer #2 (Remarks to the Author):

The authors have sufficiently addressed my concerns in the revised manuscript and rebuttal. Additional analysis and experimental data provided strengthen their conclusions. Although the perturbation experiments would have further helped confirm the lineage potential of Sox1+Foxa2+ state's differentiation potential, given the scope of the study, the authors rationale for not performing perturbation experiments is acceptable for this reviewer.

Reviewer #2 (Remarks on code availability):

I do not have sufficient expertise to review the code.

Reviewer #3 (Remarks to the Author):

Just some thoughts, that do not require an answer:

4. On the topic of the Figure 1d and removal of certain features from the analysis: of course it is always good to start with a full set of features to see the big picture. However, narrowing down the set of features to a subset of interest can reveal further subpopulations. This is related to "variable trimming" techniques in regression and clustering analysis.

8. Sure, Fisher's Exact Test works, mine was just an example

I want to thank the authors for the answers and the adjustments to the manuscript, I do not have many further comments as I find this paper publication-ready.

REVIEWERS' COMMENTS

Reviewer #1 (Remarks to the Author):

The authors have presented compelling new data demonstrating the existence of a SOX2+ FOXA2+ population in the embryo in vivo, with RNA sequencing providing a more in depth characterization of these cells. This significantly enhances the impact of their findings.

Thank you very much for your time and feedback!

However, there are still 2 major points that should be addressed. Most importantly, the authors still do not provide evidence that this population functions as a bi-potent progenitor. Establishing such a definition requires single-cell functional experiments and lineage tracing, neither of which are included in this study. Consequently, any references to these cells as bi-potent progenitors should be omitted, although the authors may speculate on this possibility in the discussion.

Thank you. Based on this reviewer's feedback and editorial suggestions, we have now avoided any definitive statements about the differentiation potential of SOX1+FOXA2+ cells. Rather, as suggested by editor, we use speculative words like 'suggests', 'possible', 'potentially indicative'.

Second, the authors frequently describe cells as "lineage committed" or "uncommitted" without supporting functional experiments. Functional assays are necessary to determine a cell's commitment to a specific lineage. While the manuscript classifies cells as "uncommitted" based on a limited set of lineage markers, it is possible that these cells express markers of other lineages not examined, making it inaccurate to label them as "uncommitted." This should also be rectified in the text with more appropriate terminology.

Thank you. Based on this reviewer's feedback and editorial suggestions, we have now changed the way we describe the cells. Rather than calling them lineage committed or uncommitted, based on editor's suggestion, we refer to the cells by their gene expression only.

Reviewer #2 (Remarks to the Author):

The authors have sufficiently addressed my concerns in the revised manuscript and rebuttal. Additional analysis and experimental data provided strengthen their conclusions. Although the perturbation experiments would have further helped confirm the lineage potential of Sox1+Foxa2+ state's differentiation potential, given the scope of the study, the authors rationale for not performing perturbation experiments is acceptable for this reviewer.

Reviewer #2 (Remarks on code availability):

I do not have sufficient expertise to review the code.

Thank you very much for your time and feedback!

Reviewer #3 (Remarks to the Author):

Just some thoughts, that do not require an answer:

4. On the topic of the Figure 1d and removal of certain features from the analysis: of course it is always good to start with a full set of features to see the big picture. However, narrowing down the set of features to a subset of interest can reveal further subpopulations. This is related to "variable trimming" techniques in regression and clustering analysis.
8. Sure, Fisher's Exact Test works, mine was just an example

I want to thank the authors for the answers and the adjustments to the manuscript, I do not have many further comments as I find this paper publication-ready.

Thank you very much for your time and feedback!